# Replication Study: Biomechanical remodeling of the microenvironment by stromal caveolin-1 favors tumor invasion and metastasis

Mee Rie Sheen[1†§], Jennifer L Fields[1†], Brian Northan[2‡], Judith Lacoste[2‡], Lay-Hong Ang[3], Steven Fiering[1], Reproducibility Project: Cancer Biology*

[1]Geisel School of Medicine at Dartmouth, Department of Microbiology and Immunology, Lebanon, United States; [2]MIA Cellavie Inc, Montreal, Canada; [3]Beth Israel Deaconess Medical Center, Harvard Medical School, Boston, United States

## REPRODUCIBILITY
PROJECT
CANCER BIOLOGY

**\*For correspondence:**
tim@cos.io;
nicole@scienceexchange.com

[†]These authors contributed equally to this work
[‡]These authors also contributed equally to this work

**Present address:** [§]Department of Surgery, Massachusetts General Hospital, Harvard Medical School, Boston, United States

**Group author details:**
Reproducibility Project: Cancer Biology See page 18

**Abstract** As part of the Reproducibility Project: Cancer Biology we published a Registered Report (Fiering et al., 2015) that described how we intended to replicate selected experiments from the paper 'Biomechanical remodeling of the microenvironment by stromal caveolin-1 favors tumor invasion and metastasis' (Goetz et al., 2011). Here we report the results. Primary mouse embryonic fibroblasts (pMEFs) expressing caveolin 1 (Cav1WT) demonstrated increased extracellular matrix remodeling *in vitro* compared to Cav1 deficient (Cav1KO) pMEFs, similar to the original study (Goetz et al., 2011). *In vivo*, we found higher levels of intratumoral stroma remodeling, determined by fibronectin fiber orientation, in tumors from cancer cells co-injected with Cav1WT pMEFs compared to cancer cells only or cancer cells plus Cav1KO pMEFs, which were in the same direction as the original study (Supplemental Figure S7C; Goetz et al., 2011), but not statistically significant. Primary tumor growth was similar between conditions, like the original study (Supplemental Figure S7Ca; Goetz et al., 2011). We found metastatic burden was similar between Cav1WT and Cav1KO pMEFs, while the original study found increased metastases with Cav1WT (Figure 7C; Goetz et al., 2011); however, the duration of our *in vivo* experiments (45 days) were much shorter than in the study by Goetz et al. (2011) (75 days). This makes it difficult to interpret the difference between the studies as it is possible that the cells required more time to manifest the difference between treatments observed by Goetz et al. We also found a statistically significant negative correlation of intratumoral remodeling with metastatic burden, while the original study found a statistically significant positive correlation (Figure 7Cd; Goetz et al., 2011), but again there were differences between the studies in terms of the duration of the metastasis studies and the imaging approaches that could have impacted the outcomes. Finally, we report meta-analyses for each result.

## Introduction

The Reproducibility Project: Cancer Biology (RP:CB) is a collaboration between the Center for Open Science and Science Exchange that seeks to address concerns about reproducibility in scientific research by conducting replications of selected experiments from a number of high-profile papers in the field of cancer biology (Errington et al., 2014). For each of these papers a Registered Report detailing the proposed experimental designs and protocols for the replications was peer reviewed and published prior to data collection. The present paper is a Replication Study that reports the results of the replication experiments detailed in the Registered Report (Fiering et al., 2015) for a

2011 paper by Goetz et al., and uses a number of approaches to compare the outcomes of the original experiments and the replications.

In 2011, Goetz et al. reported that Caveolin-1 (Cav1), an activator of Rho/ROCK signaling (*Joshi et al., 2008*), remodels the intratumoral microenvironment facilitating tumor invasion and correlating with increased metastatic burden. By regulating the Rho inhibitor p190RhoGAP, Cav1 expression results in cancer-associated fibroblasts (CAFs) that promote extracellular matrix (ECM) alignment and stiffening (*Goetz et al., 2011*). As the ECM is stiffened, it may direct cancer cell invasion into the surrounding stroma for eventual metastasis (*Wang et al., 2016*). To specifically address the role of Cav1 in the tumor stroma, primary mouse embryonic fibroblasts (pMEFs) derived from either wild-type (Cav1WT) or Cav1 knockout (Cav1KO) mice were co-injected with LM-4175 tumor cells, a cell line of a lung metastasis derived MDA-MB-231 breast cancer cells (*Minn et al., 2005*). The number of metastases was increased when Cav1 was present compared to the Cav1KO condition (*Goetz et al., 2011*). Additionally, there was increased ECM remodeling (e.g. fibronectin fiber alignment) of the primary tumors when Cav1WT pMEFs were present compared to Cav1KO pMEFs (*Goetz et al., 2011*). Intratumoral fibronectin alignment was correlated with increased metastatic burden suggesting Cav1 positive stroma are permissive for tumor progression (*Goetz et al., 2011*).

The Registered Report for the paper by Goetz et al. described the experiments to be replicated (Figure 7C and Supplemental Figures S2A and S7C), and summarized the current evidence for these findings (*Fiering et al., 2015*). Since that publication additional studies have reported that fibronectin assembly by CAFs stimulate cancer cell invasion (*Attieh et al., 2017*). Several studies have also reported the correlation of stromal Cav1 expression and clinical outcome, with some associating high expression of Cav1 with unfavorable outcome (*Chatterjee et al., 2015*; *Sun et al., 2017*) and some associating Cav1 expression with favorable clinical outcome (*Eliyatkin et al., 2018*; *Neofytou et al., 2017*). The reported tumor-promoting and tumor-suppressive functions of Cav1 are likely due to cell-specific effects, physiological context, and cancer stage (*Celus et al., 2017*).

The outcome measures reported in this Replication Study will be aggregated with those from the other Replication Studies to create a dataset that will be examined to provide evidence about reproducibility of cancer biology research, and to identify factors that influence reproducibility more generally.

## Results and discussion

### Isolation and characterization of Cav1 wild-type and Cav1 knockout primary MEFs

To test the effect of Cav1 in the tumor stroma, we isolated Cav1WT and Cav1KO pMEFs. The experimental approach to isolate and characterize the pMEFs was described in Protocol 1 and 2 of the Registered Report (*Fiering et al., 2015*). Isolated pMEFs were assessed for Smooth Muscle Actin (SMA) expression to determine if Cav1WT pMEFs had increased expression compared to Cav1KO pMEFs. This was suggested during peer review of the Registered Report as a marker of increased fibroblast activation and ECM remodeling capabilities of the pMEFs (*Fiering et al., 2015*). We observed similar SMA expression between Cav1WT and Cav1KO pMEFs (*Figure 1A,B*). The pMEFs were used within a few passages after isolation, however, growth on a stiff substrate (i.e. plastic) can lead to increased SMA expression (*Jones and Ehrlich, 2011*; *Shi et al., 2013*), which could have masked any subtle differences in expression between Cav1WT and Cav1KO pMEFs. Indeed, the original study observed that three-dimensional (3D) growth preferentially raised SMA expression in Cav1WT immortalized MEFs close to levels of SMA in pMEFs grown under two-dimensional (2D) conditions (*Goetz et al., 2011*). Instead, we performed a collagen contraction assay to test if Cav1WT pMEFs have increased ECM remodeling capabilities compared to Cav1KO pMEFs. This was reported for immortalized MEFs in the original study; however, for pMEFs the data were 'not shown' in the published paper because of the journal policy at *Cell* restricting the number of supplemental figures allowed (del Pozo, personal communication). Although the data were not reported, the original study stated that the ECM remodeling capabilities of Cav1KO pMEFs were reduced compared to Cav1WT pMEFs, similar to the results reported with immortalized MEFs (*Goetz et al., 2011*). In

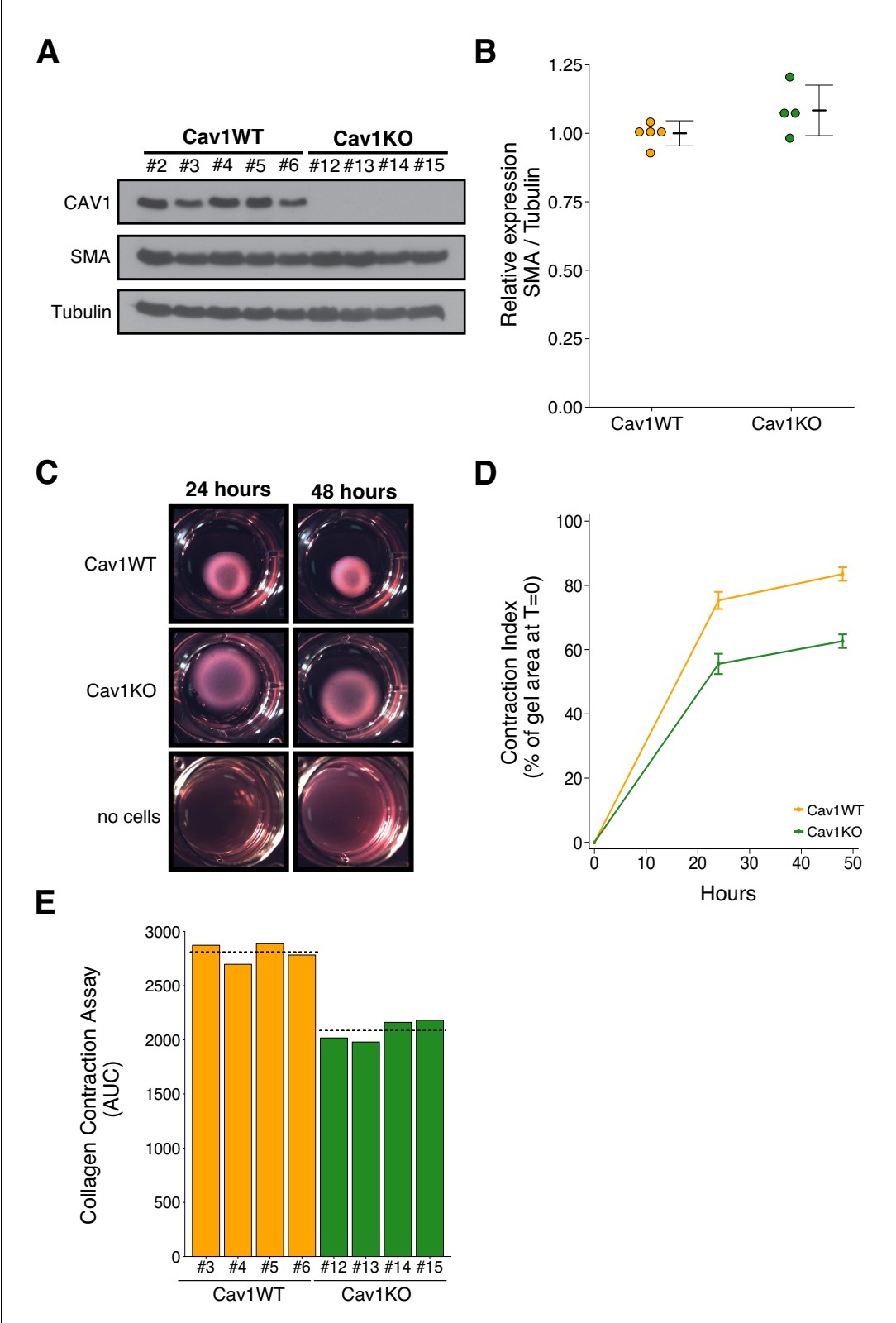

**Figure 1.** Characterization of Cav1 wild-type and Cav1 knockout pMEFs. Primary MEFs (pMEFs) from wild-type (WT) or knockout (KO) embryos were examined for increased fibroblast activation and extracellular matrix (ECM) remodeling capabilities *in vitro*. (**A**) Western blots of the indicated pMEFs probed with antibodies against caveolin-1 (CAV1), alpha-smooth muscle actin (SMA), and gamma-Tubulin. Numbers indicate individual pMEF clones. (**B**) Western blot bands were quantified, SMA levels were normalized to Tubulin, and protein expression are presented relative to Cav1WT. Dot plot

*Figure 1 continued on next page*

*Figure 1 continued*

with means reported as crossbars and error bars represent *SD*. Number of individual clones per group: Cav1WT = 5, Cav1KO = 4. Exploratory analysis: Student's two-tailed *t*-test; *t*(7) = 1.791, *p*=0.116; Cohen's *d* = 1.20, 95% CI [−0.52, 2.92]. (C) Representative images from collagen contraction assay of the indicated conditions at 24 or 48 hr after plating. (D) Line graph of contraction index, measured as the change in percent of gel area at time of plating, of Cav1WT and Cav1KO pMEFs at the indicated times after plating. Means reported and error bars represent *SD*. Number of individual clones per group: Cav1WT = 4, Cav1KO = 4. (E) The contraction index was used to calculate the area under the curve (AUC) for each clone. Bar plots for each individual clone tested (numbers indicate same clone number as in A). Dashed lines indicate means of each group. Exploratory analysis: Student's two-tailed *t*-test; *t*(6) = 10.80, *p*=$3.72\times10^{-5}$; Cohen's *d* = 7.64, 95% CI [2.66, 12.62]. Additional details for this experiment can be found at https://osf.io/na5h2/.

this study, we also found Cav1KO pMEFs had decreased contraction compared to Cav1WT pMEFs (*Figure 1C–E*). This result is consistent with Cav1 contributing to fibroblast contractility. To summarize, we were unable to observe differences in SMA expression between Cav1WT and Cav1KO pMEFs in 2D conditions on a rigid substrate, but did observe contraction in Cav1WT pMEFs, that was reduced in Cav1KO pMEFs, a result that was in the same direction as the original study.

## Subcutaneous tumorigenicity assay of tumor cells co-injected with Cav1WT or Cav1KO primary MEFs

We next used the pMEFs to replicate an experiment to test whether stromal Cav1 remodels the intratumoral microenvironment and facilitates tumor cell metastasis. This experiment is similar to what was reported in Figure 7C and Supplemental Figure 7C of *Goetz et al. (2011)* and described in Protocols 3 and 4 in the Registered Report (*Fiering et al., 2015*). Tumor cells engineered to express luciferase (LM-4175) were mixed with Cav1WT or Cav1KO pMEFs, or not mixed with pMEFs (control group), and injected subcutaneously into female nude mice. While the original study also tested the role of p190RhoGAP by injecting LM-4175 cells mixed with p190RhoGAP-silenced Cav1KO pMEFs, this replication attempted did not attempt to include this condition. To determine the experimental endpoint, we first performed a pilot experiment and established that mice should be euthanized 45 days after cell injection to maximize the length of time for tumor growth while minimizing animal suffering. Importantly, while the original study did not report tumor sizes, in this pilot experiment tumor burden was determined to be excessive (1.5 cm$^3$) at an experimental endpoint that was 25 days shorter than the original study, which maintained mice for 70 days after injection. Thus, it is possible that tumors grew more rapidly in this replication attempt than the original study. Following the same time course as the pilot experiment we injected female nude mice with LM-4175 cells with or without Cav1WT or Cav1KO pMEFs. Similar to the pilot study we observed criteria warranting euthanasia in some mice (e.g. ulceration at the tumor site) confirming that 45 days after injection was the appropriate endpoint. Before euthanasia, each mouse was injected with luciferin to monitor primary tumor growth and metastasis formation. We found there was not a statistically significant difference in primary tumor growth between the three groups (Kruskal-Wallis: H(2) = 0.0439, *p*=0.978), with a median (*Mdn*) bioluminescence of 1.92 × 109 photons/sec [n = 10] for LM-4175, 1.92 × $10^9$ photons/sec [n = 26] for LM-4175 plus Cav1WT pMEFS, and 1.64 × $10^9$ photons/sec [n = 25] for LM-4175 plus Cav1KO pMEFs (*Figure 2A,B*). This compares to a *Mdn* bioluminescence of 2.16 × $10^{10}$ photons/sec [n = 6] for LM-4175, 1.54 × $10^{10}$ photons/sec [n = 13] for LM-4175 plus Cav1WT pMEFS, and 2.08 × $10^{10}$ photons/sec [n = 15] for LM-4175 plus Cav1KO pMEFs reported in the original study (*Goetz et al., 2011*).

To assess metastatic burden, we excised the same organs examined in the original study and reimaged them *ex vivo*. Similar to the original study, we observed that the incidence of a metastatic foci, when considering all the mice examined, was highest in the lymph node (RP:CB: 41% (25 out of 61 mice); *Goetz et al., 2011*: 77% (27 out of 35)). In this replication attempt we observed the lowest incidence of metastatic foci in the kidney and liver (13% (8 out of 61)) while the original study observed the lowest incidence in the spleen (37% (13 out of 35)). When considering the total number of metastatic foci detected among all the examined organs, we found mice injected with LM-4175 cells formed a median of 0 metastatic foci (range: 0–115; incidence: 40% (4 out of 10 mice)), mice injected with LM-4175 cells plus Cav1WT pMEFS formed a median of 3 metastatic foci (range: 0–25; incidence: 62% (16 out of 26)), and mice injected with LM-4175 cells plus Cav1KO pMEFs formed a median of 5 metastatic foci (range: 0–33; incidence: 84% (21 out of 25)) (*Figure 2A,C*). The original

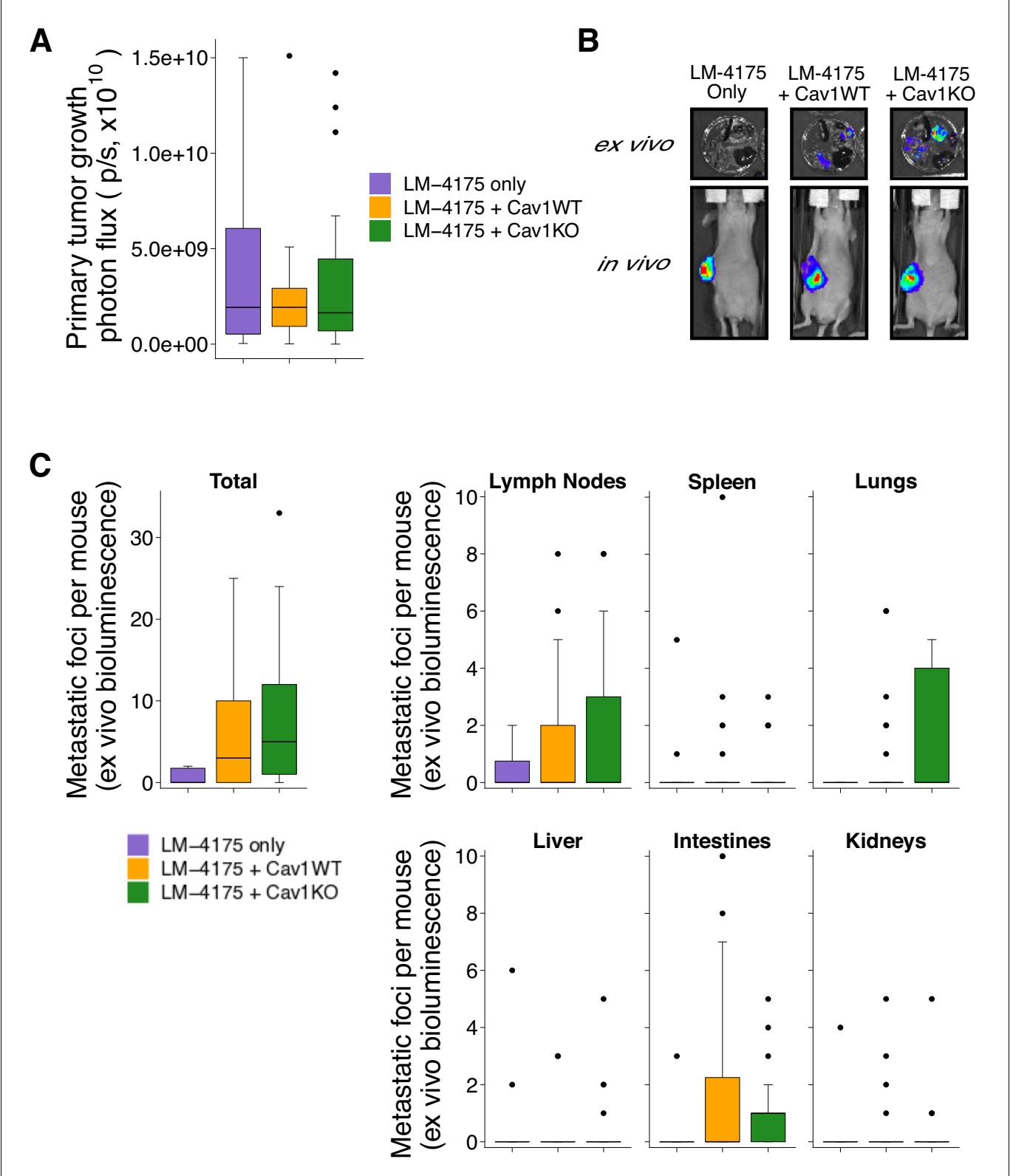

**Figure 2.** Primary tumor growth and metastatic burden from subcutaneous tumorigenicity assay. Female nude mice were subcutaneously injected with $1 \times 10^6$ LM-4175 cells mixed with or without $1 \times 10^6$ Cav1WT or Cav1KO pMEFs and monitored for 45 days. (**A**) At the end of the experiment primary tumors were imaged *in vivo*. Box and whisker plot of primary tumor photon flux with median represented as the line through the box and whiskers representing values within 1.5 IQR of the first and third quartile. Number of primary mice per group: LM-4175 only (control group)=10, LM-4175 plus

*Figure 2 continued on next page*

*Figure 2 continued*

Cav1WT pMEFs = 26, LM-4175 plus Cav1KO pMEFs = 25. Kruskal-Wallis test on all three groups: H(2) = 0.0439, *p*=0.978. (B) Representative images of primary tumors *in vivo* and extracted organs *ex vivo*. (C) The indicated organs were dissected, imaged *ex vivo*, and individual metastatic foci were blindly quantified. Box and whisker plots of metastatic foci counts for each organ and total metastatic counts with median represented as the line through the box and whiskers representing values within 1.5 IQR of the first and third quartile (dots represent outliers). Note: the y-axes have been truncated for visualization purposes and excludes two outliers from Total (LM-4175 only), one from Lymph Nodes (LM-4175 plus Cav1WT), seven from Lung (two from LM-4175 only; five from LM-4175 plus Cav1KO), and one from Intestines (LM-4175 only). The excluded outliers were included in the statistical analysis below. Number of mice per group: LM-4175 only = 10, LM-4175 plus Cav1WT pMEFs = 26, LM-4175 plus Cav1KO pMEFs = 25. Planned Wilcoxon-Mann-Whitney comparison on total metastatic counts between LM-4175 only and LM-4175 plus Cav1WT pMEFs: *U* = 103, uncorrected *p*=0.318 with *a priori* alpha level of 0.0167, Bonferroni corrected *p*=0.954, Cliff's delta = 0.21, 95% CI [−0.08, 0.46]. Planned Wilcoxon-Mann-Whitney comparison on total metastatic counts between LM-4175 only and LM-4175 plus Cav1KO pMEFs: *U* = 175.5, uncorrected *p*=0.062, Bonferroni corrected *p*=0.185, Cliff's delta = 0.40, 95% CI [0.10, 0.64]. Planned Wilcoxon-Mann-Whitney comparison on total metastatic counts between LM-4175 plus Cav1WT pMEFs and LM-4175 plus Cav1KO pMEFs: *U* = 389.5, uncorrected *p*=0.219, Bonferroni corrected *p*=0.657, Cliff's delta = −0.20, 95% CI [−0.47, 0.11]. Additional details for this experiment can be found at https://osf.io/bq54u/.

study reported a median of 3.5 (range: 0–4; incidence: 67% (4 out of 6 mice)) for LM-4175, 26 (range: 1–67; incidence: 100% (12 out of 12)) for LM-4175 plus Cav1WT pMEFs, and 8 (range: 0–34; incidence: 93% (14 out of 15)) for LM-4175 plus Cav1KO pMEFs (*Goetz et al., 2011*). There are multiple approaches that could be taken to explore these data; however, to provide a direct comparison to the original data, we conducted the analysis specified *a priori* in the Registered Report (*Fiering et al., 2015*). To test if the number of metastatic foci differed between the three groups we performed three planned comparisons, which were not statistically significant (see *Figure 2* figure legend). Interpretation of the metastatic burden should take into consideration the shorter time from cell injection until euthanasia conducted in this replication attempt, which was 25 days (36%) shorter than the original study. To summarize, for assessment of metastasis formation we found results that were not in the same direction as the original study and not statistically significant.

There are a number of factors that can affect the evaluation of tumor growth and metastasis formation using bioluminescence imaging. For *in vivo* imaging, the depth and location of the tumor, as well as the thickness or color of the animal's skin can alter the bioluminescent signal (*Baba et al., 2007*). The type of anesthetics used can impact the luciferase reaction (*Keyaerts et al., 2012*) as well as the route of injection of D-luciferin. So while both studies used an intraperitoneal injection there can be variation in the signal due to changes in the rate of absorption across the peritoneum (*Close et al., 2010*). Thus, intravenous and subcutaneous administration of D-luciferin have been suggested alternatives (*Keyaerts et al., 2008*; *Khalil et al., 2013*). The imaging time postinjection can also affect the sensitivity of the bioluminescent signal as well as differences in instrumentation settings (*Burgos et al., 2003*; *Rettig et al., 2006*). Additionally, the animal diet can also affect the background gut phosphorescence with standard mouse chow with plant material displaying greater phosphorescence compared to a diet without plant material (*Zinn et al., 2008*). Finally, an immune response against luciferase has also been reported to restrict tumor growth and metastatic potential of luciferase expressing tumor cells (*Baklaushev et al., 2017*).

As noted above, the difference in experimental timing could have had important effects on both the extent and patterns of metastases observed. There are numerous cellular processes that tumor cells must accomplish to form metastases, including evasion of immune responses and programmed cell death, invasion of the host stroma, escape through vasculature and/or lymphatics, and survival and growth in distant sites (*Chambers et al., 2002*). Thus, there are multiple steps during malignant progression that are influenced by a number of factors, particularly time. Most experimental systems, however, do not model all of the steps necessary for metastasis formation (*Saxena and Christofori, 2013*). Subcutaneous approaches, such as the model used in the original study and this replication, can robustly model *in vivo* tumor growth, as well as local invasion toward skin mesenchyme, but do not reliably recapitulate metastatic behavior, likely because of ectopic anatomical context (*Antonello and Nucera, 2014*; *Pearson and Pouliot, 2013*). Experimental timing of primary tumor growth and spontaneous metastasis are important to maintain to minimize confounding variables especially since growth is nonlinear (*Tyuryumina and Neznanov, 2018*). This can be complicated when primary tumor growth necessitates the sacrifice of animals before sufficient time for metastatic development. Monitoring and reporting tumor growth, such as tumor volumes for each animal at

the experimental endpoint, can allow for mitigation strategies if there are variations in the growth of tumors between studies. For example, the primary tumor could be resected at a specific time, or tumor size, to allow for a longer follow-up of metastasis development. Additionally, other growth monitoring criteria, such as using biomarkers to visualize metastatic burden *in vivo* without sacrificing animals, should be considered in the experimental design of future studies.

## Intratumoral stroma remodeling

In addition to monitoring metastasis formation we blindly examined intratumoral stroma remodeling in a random subset of the primary tumors. Tumors sections were stained for fibronectin and SMA using the same antibodies and protocol as the original study. Fibronectin staining gave specific staining with little background in control conditions (*Figure 3—figure supplement 1A*); however, there was high non-specific staining observed with SMA (*Figure 3—figure supplement 1B*). In an attempt to reduce the non-specific staining we included a mouse-on-mouse blocking step since a mouse anti-SMA antibody was being used on mouse tissue. While this reduced background staining, we observed heterogeneity in the patterns (e.g. fibrillar structures, bright dots) and intensities within the tumors (*Figure 3—figure supplement 1C*). This introduced an unanticipated difficulty in needing to separate out the bright dots, which appeared specific based on the controls, from the fibrous SMA. As such, we did not conduct the SMA analysis that was outlined in the Registered Report.

We next quantified intratumoral orientation of fibronectin fibers from 10 random images per tumor. As specified in the Registered Report (*Fiering et al., 2015*), we attempted to determine fibronectin orientation using MetaMorph software as described in the original study, but found that the Integrated Morphometry Analysis (IMA) function to reveal objects of interest was unable to be executed as there were too many objects to process (see detailed approach in Materials and methods). Instead, we created a workflow using the KNIME analytics platform (*Berthold et al., 2007*) that allows the integration of ImageJ commands into a single workflow to ensure all images of the dataset are processed in an identical manner. We were unable to perform the exact methodology since there were thousands of objects remaining in control conditions after performing the 35% threshold at the maximum internal intensity as prespecified in the Registered Report. The number of objects detected at this step was higher than the number MetaMorph IMA function could handle. A large portion of these objects were very small, therefore we included an additional parameter that selected objects that were above a certain size (*Figure 3—figure supplement 2A*). The fibronectin fiber orientation among the various images was determined by arbitrarily setting the mode angle, that represents the angle with the most fibers observed, to 0° for each image and then calculating the average percentage of fibers oriented within 20° of the mode angle (i.e. −20° to 20°) (*Amatangelo et al., 2005*). We found the percentage of fibers within 20° of the mode was highest in tumors from LM-4175 plus Cav1WT pMEFs [$Mdn$ = 44.1%, interquartile range (IQR) = 41.3–45.2%, n = 8] or LM-4175 cells [$Mdn$ = 44.6%, IQR = 38.0–46.0%, n = 5] compared to tumors from LM-4175 plus Cav1KO pMEFs [$Mdn$ = 41.2%, IQR = 40.0–41.8%, n = 7] (*Figure 3A,B*, *Figure 3—figure supplement 3A,B*). To test if the orientation of fibronectin fibers differed we performed the two planned comparisons outlined in the Registered Report (LM-4175 vs LM-4175 plus Cav1WT pMEFs; LM-4175 plus Cav1WT pMEFs vs LM-4175 plus Cav1KO pMEFs), which were not statistically significant (see *Figure 3* figure legend). This compares to the original study that reported a statistically significant increase in fibronectin fiber alignment when LM-4175 cells were co-injected with Cav1WT pMEFs [$Mdn$ = 52.5%, IQR = 43.6–55.4%, n = 8] compared to LM-4175 cells [$Mdn$ = 36.9%, IQR = 36.8–37.7%, n = 5] or LM-4175 plus Cav1KO pMEFs [$Mdn$ = 42.2%, IQR = 40.8–43.2%, n = 10], suggesting stromal Cav1 remodels the intratumoral microenvironment (*Goetz et al., 2011*). To summarize, we found results that were in the same direction as the original study and not statistically significant where predicted. Interpretation of these results should take into consideration the changes in analysis workflow between the original and replication studies. It is unknown what the impact of the change in methods are since the workflow used for the original study could not be implemented on the replication data and vice versa. Although, despite these differences, the median value of percent of fibronectin fibers oriented within 20% across all tumors was similar between this replication attempt [$Mdn$ = 42.1%, IQR = 39.1–44.7%, n = 20] and the original study [$Mdn$ = 42.5%, IQR = 37.9–45.1%, n = 23]. Nonetheless, the fibrillar nature of fibronectin staining might not be fully captured due to variations in staining and imaging (e.g. image noise), a common challenge that affects the quality of fluorescence-based images because of the low-light nature of the signal. It was

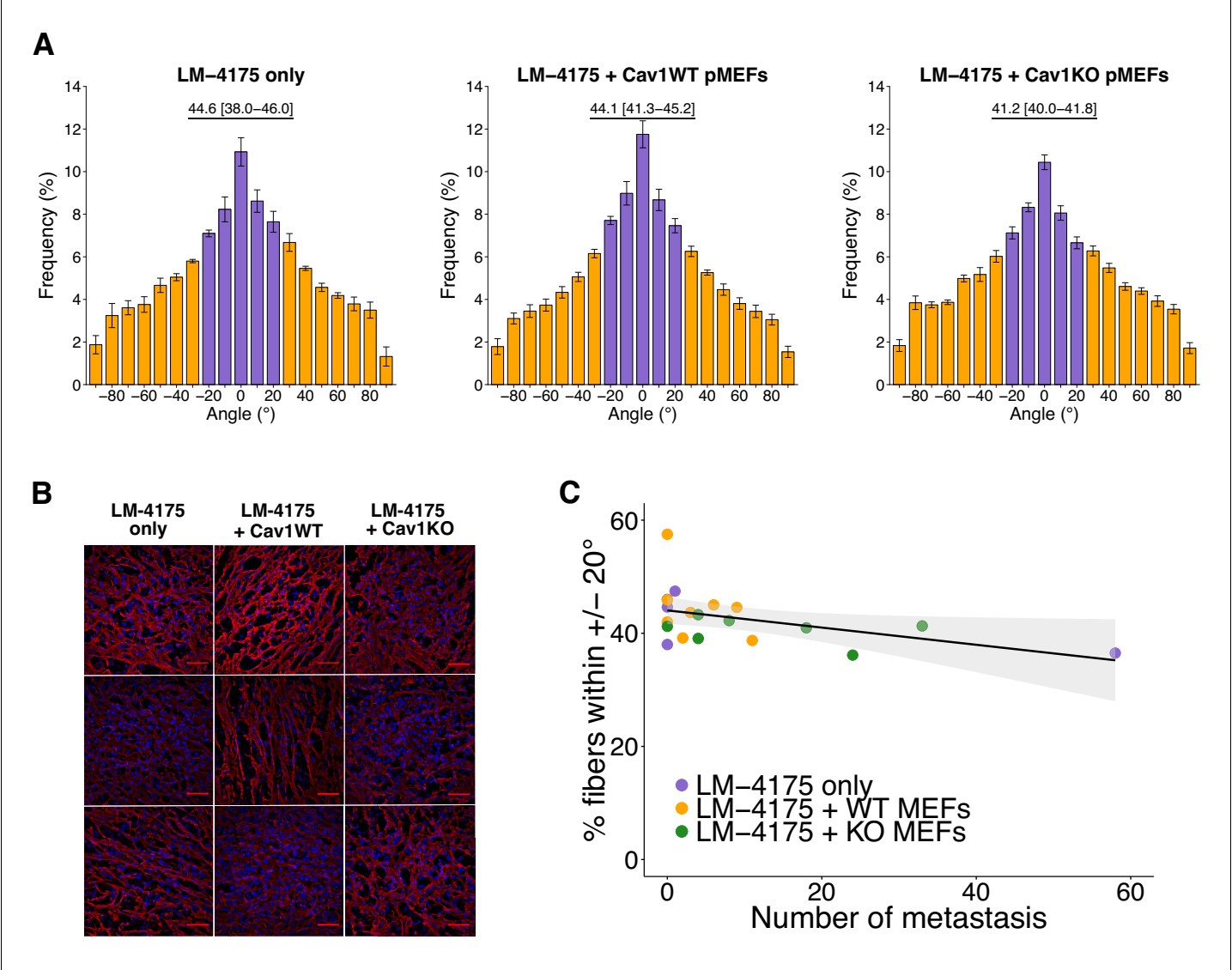

**Figure 3.** Intratumoral fibronectin fiber orientation and correlation to metastasis. A random subset of the primary tumors from the subcutaneous tumorigenicity assay (20 of 61 mice) were stained for fibronectin and analyzed to determine the average percentage of fibers oriented within 20° of the modal angle. (A) Bar graphs of the frequency of fibronectin fiber angle plotted relative to the modal angle (set at 0°). Means reported and error bars represent s.e.m. Number of mice, and thus tumors, per group: LM-4175 only = 5, LM-4175 plus Cav1WT pMEFs = 8, LM-4175 plus Cav1KO pMEFs = 7. Values reported above bar graphs indicate the median and interquartile range (IQR) of percent of fibers oriented within 20° of the modal angle (−20° to 20°, represented as purple bars) for each tumor. Planned Wilcoxon-Mann-Whitney comparison on percent of fibers oriented within 20° of the modal angle between LM-4175 only and LM-4175 plus Cav1WT pMEFs: $U$ = 19, uncorrected $p$=0.884 with *a priori* alpha level of 0.025, Bonferroni corrected $p$>0.99. Planned Wilcoxon-Mann-Whitney comparison on percent of fibers oriented within 20° of the modal angle between LM-4175 plus Cav1WT pMEFs and LM-4175 plus Cav1KO pMEFs: $U$ = 13, uncorrected $p$=0.0826, Bonferroni corrected $p$=0.165. (B) Different fields of views (fov) of the immunostained primary tumors for fibronectin and Hoechst. Three independent tumors derived from LM4175 only (first column, top to bottom: tumor 2 fov5, tumor 4 fov9, tumor 34 fov3), LM-4175 plus Cav1WT pMEFs (second column, top to bottom: tumor 16 fov7, tumor 51 fov 7, tumor 61 fov2), and LM-4175 plus Cav1KO pMEFs (third column, top to bottom: tumor 20 fov4, tumor 30 fov1, tumor 47 fov6). Scale bar: 50 μm. Fibronectin signal is pseudo-colored in red (microscope emission peak wavelength: 614 nm) and Hoechst signal is pseudo-colored in blue (microscope peak emission wavelength: 454 nm). Images are maximum intensity projections of the Z-stacks, corrected for background (as described in Materials and methods - Fibronectin fiber analysis) and displayed in the same range of grey levels. (C) Scatter plot of percentage of fibers within 20° of the modal angle and total number of metastatic counts for 20 tumors analyzed for fibronectin orientation. Line represents spearman rank correlation and light gray region represents 95% CI. Spearman rank-order correlation analysis: $r_s$(18) = −0.50, $p$=0.025. Additional details for this experiment can be found at https://osf.io/bq54u/.

The online version of this article includes the following figure supplement(s) for figure 3:

**Figure supplement 1.** Immunostaining of primary tumors.

*Figure 3 continued on next page*

**Figure supplement 2.** Additional measures of fibronectin fiber analysis.
**Figure supplement 3.** Fields of view from fibronectin fiber analysis.
**Figure supplement 4.** KNIME workflow.

unclear if this occurred in the original study, and if so, what was performed to manage this. Others have suggested methods to evaluate noise (*Heintzmann et al., 2018*; *Murray, 2007*), with management steps implemented in either the equipment or in the image process and analysis protocol. Additionally, data analysis should be done blinded to conditions and batch processed, with specific details of what will occur stated prior to data collection, such as in a pre-registered analysis plan, to minimize confirmation bias (*Wagenmakers et al., 2012*).

We also explored additional methods to examine fiber orientation. Anisotropy, a measure of orderly structure, was measured with FibrilTool (*Boudaoud et al., 2014*; proposed by the original authors during preparation of the Registered Report), coherency, the degree to which the local features are oriented, was measured with OrientationJ (*Rezakhaniha et al., 2012*), and a blinded manual scoring was performed to assess the frequency of parallel fibers. These additional measures were found to be well correlated with the percentage of fibers oriented within 20° of the mode angle (*Figure 3—figure supplement 2B*). The same statistical comparisons between the three groups that were performed above were also explored, which gave similar results (*Figure 3—figure supplement 2C*). Although the full range of possible methods were not explored, these concordant results indicate the robustness of the findings (*Silberzahn et al., 2018*; *Steegen et al., 2016*).

Finally, intratumoral fibronectin fiber alignment was examined to determine if there was a correlation with metastasis formation. Results of the Spearman's rank-order correlation indicated that there was a statistically significant negative relationship between the number of metastatic foci and percentage of fibers within 20° of the mode ($r_s(18) = -0.50$, p=0.025) (*Figure 3C*). The same type of analysis was reported in the original study, which indicated a statistically significant positive relationship (*Goetz et al., 2011*). Interpretation of this analysis should take into consideration the results above, especially since the shorter experimental timing could have impacted the number of metastases observed. Additionally, the primary tumors and metastatic foci counts used for the correlation analysis were a random subset of all the mice evaluated in this study. To summarize, we found results that were statistically significant in the opposite direction as the original study.

Interpretation of the above results should take into account experimental differences between the original and replication studies. The decreased experimental endpoint (45 days instead of 70 days) could have had important effects on both the extent and patterns of metastases as well as affect the fibronectin pattern associated with intratumoral remodeling. That is, the shorter *in vivo* experimental timing might not have allowed for the same level of metastatic progression to occur in this replication compared to the original study. Additionally, the *in vivo* ECM remodeling capabilities of the primary fibroblasts used in this study are unknown due to cross reaction during SMA staining despite using the same protocol as the original study. Thus, while the *in vitro* contractility assay observed a difference between the pMEFs, a larger difference might be required to observe an effect on intratumoral orientation with this experimental design. An examination of the involvement of p190RhoGAP should also be considered in the experimental design of future studies. Importantly, observing different outcomes with similar experimental designs are informative to establish the range of conditions under which a given effect can be observed (*Bailoo et al., 2014*).

## Meta-analyses of original and replication effects

We performed a meta-analysis using a random-effects model, where possible, to combine each of the effects described above as pre-specified in the confirmatory analysis plan (*Fiering et al., 2015*). To provide a standardized measure of the effect, a common effect size was calculated for each effect from the original and replication studies. Cliff's delta (*d*) is a non-parametric estimate of effect size that measures how often a value in one group is larger than the values from another group, while the effect size *r* is a standardized measure of the correlation (strength and direction) of the association between two variables. The estimate of the effect size of one study, as well as the associated uncertainty (i.e. confidence interval), compared to the effect size of the other study provides one

approach to compare the original and replication results (*Errington et al., 2014*; *Valentine et al., 2011*). Importantly, the width of the confidence interval (CI) for each study is a reflection of not only the confidence level (e.g. 95%), but also variability of the sample (e.g. *SD*) and sample size.

The comparisons of the primary tumor growth between the three groups of mice, LM-4175 cells injected with or without Cav1WT or Cav1KO pMEFs, which were reported in *Figure 2A* of this study and Supplemental Figure 7Ca of *Goetz et al. (2011)*, were in the same directions between the two studies and the effect size point estimate of each study was within the CI of the other study (*Figure 4A*). Furthermore, the meta-analysis was not statistically significant (*p*=0.467), suggesting primary tumor growth does not change when Cav1 is absent in the tumor stroma.

There were three comparisons of total metastasis formation between the three groups, which was reported in *Figure 2C* of this study and Figure 7Cb of *Goetz et al. (2011)*. The meta-analyses were not statistically significant for the LM-4175 vs LM-4175 plus Cav1WT pMEFs comparison (*p*=0.107) and the LM-4175 plus Cav1WT pMEFs vs LM-4175 plus Cav1KO pMEFs comparison (*p*=0.680), but was for the LM-4175 vs LM-4175 plus Cav1KO pMEFs comparison (*p*=0.0027) (*Figure 4B*). The direction of the LM-4175 vs LM-4175 plus Cav1KO pMEFs comparison was the

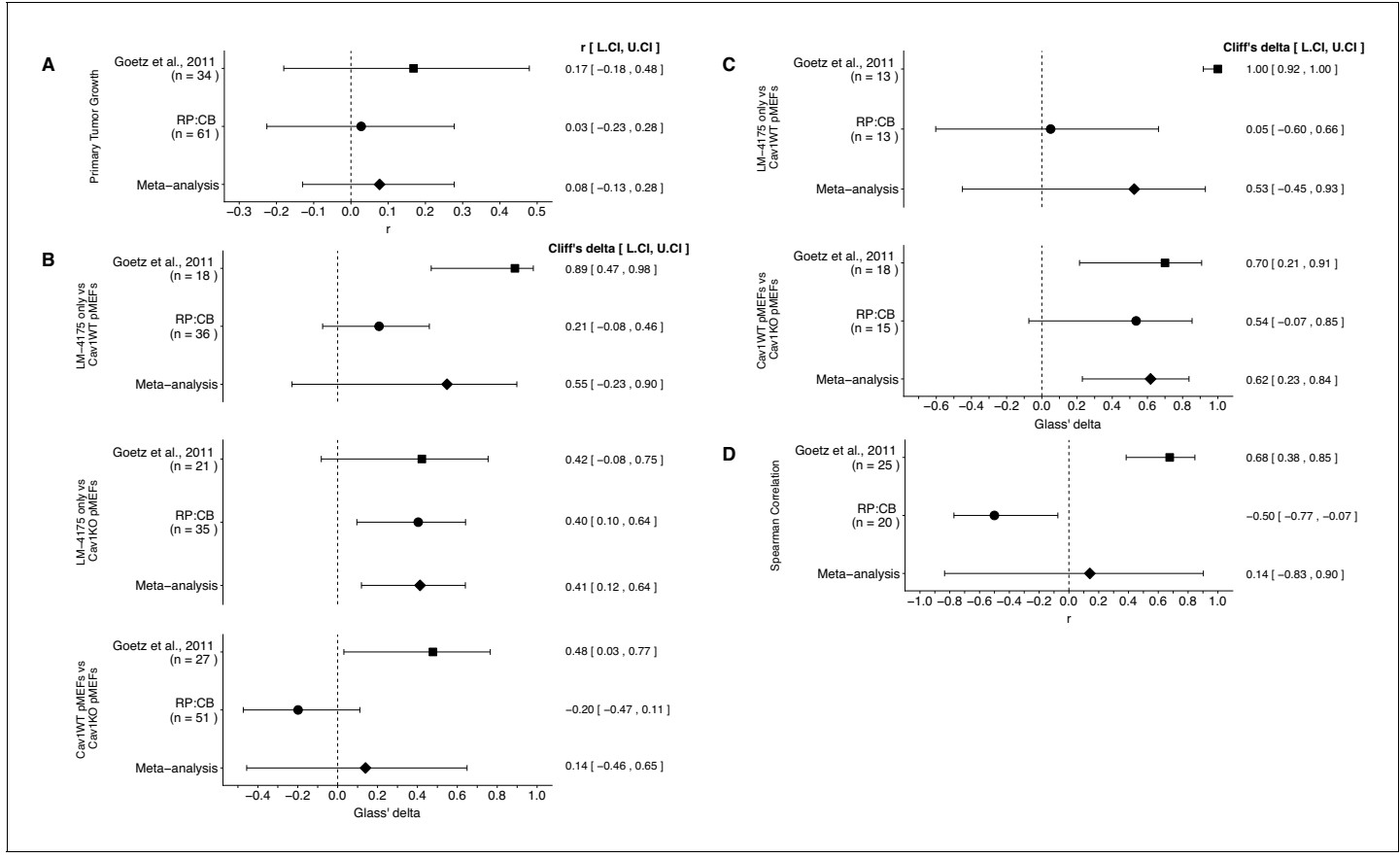

**Figure 4.** Meta-analyses of each effect. Effect size and 95% confidence interval are presented for *Goetz et al. (2011)*, this replication attempt (RP:CB), and a random effects meta-analysis to combine the two effects. The effect size *r* is a standardized measure of the correlation (strength and direction) of the association between two variables and Cliff's delta is a standardized measure of how often a value in one group is larger than the values from another group. Sample sizes used in *Goetz et al. (2011)* and this replication attempt are reported under the study name. (**A**) Primary tumor growth between mice injected with LM-4175 cells with or without Cav1WT or Cav1KO pMEFs (meta-analysis *p*=0.467). (**B**) Total metastasis counts between mice injected with LM-4175 cells and LM-4175 plus Cav1WT pMEFs (meta-analysis *p*=0.107), mice injected with LM-4175 cells and LM-4175 plus Cav1KO pMEFs (meta-analysis *p*=0.0027), and mice injected with LM-4175 plus Cav1WT pMEFs and LM-4175 plus Cav1KO pMEFs (meta-analysis *p*=0.680). (**C**) Fibronectin fiber orientation (average percentage of fibers oriented within 20° of the mode angle) between tumors from mice injected with injected with LM-4175 cells and LM-4175 plus Cav1WT pMEFs (meta-analysis *p*=0.269) and mice injected with LM-4175 plus Cav1WT pMEFs and LM-4175 plus Cav1KO pMEFs (meta-analysis *p*=1.11×10$^{-4}$). (**D**) Rank-order correlation between fibronectin fiber orientation and total metastasis counts (meta-analysis *p*=0.837). Additional details for these meta-analyses can be found at https://osf.io/rvf57/.

same in both the original study and this replication attempt with the CI of each study encompassing the effect size point estimate of the other study. The effect size point estimates of the LM-4175 vs LM-4175 plus Cav1WT pMEFs and LM-4175 plus Cav1WT pMEFs vs LM-4175 plus Cav1KO pMEFs comparisons for each study, however, were not within the CI of the other study. Additionally, for these two effects, the large CI of the meta-analyses along with statistically significant Cochran's $Q$ tests (LM-4175 vs LM-4175 plus Cav1WT pMEFs, $p=1.54\times10^{-4}$; LM-4175 plus Cav1WT pMEFs vs LM-4175 plus Cav1KO pMEFs, $p=0.0077$) suggests heterogeneity between the studies.

There were two comparisons of fibronectin fiber orientation, which was reported in *Figure 3A* of this study and Supplemental Figure 7Cc of *Goetz et al. (2011)*. Both comparisons were consistent when considering the direction of the effect; however, results varied as to whether the effect size point estimate of one study fell within the CI of the other study (*Figure 4C*). The meta-analysis for the LM-4175 plus Cav1WT pMEFs vs LM-4175 plus Cav1KO pMEFs comparison was statistically significant ($p=1.11\times10^{-4}$), which suggests Cav1 expression is necessary for intratumoral fibronectin remodeling; however, the meta-analysis for the LM-4175 vs LM-4175 plus Cav1WT pMEFs comparison was not statistically significant ($p=0.269$), which along with a statistically significant Cochran's $Q$ test ($p=0.023$) suggests heterogeneity between the studies.

Finally, the rank-order correlation that was determined in both studies to determine the association of fibronectin fiber orientation and total metastasis for all three groups, LM-4175 cells injected with or without Cav1WT or Cav1KO pMEFs, reported in *Figure 3C* of this study and Figure 7Cd of *Goetz et al. (2011)*, were not consistent when considering direction of the effect (*Figure 4D*). Furthermore, the meta-analysis was not statistically significant ($p=0.837$) with a large CI and a statistically significant Cochran's $Q$ test ($p=2.13\times10^{-5}$) that suggests heterogeneity between the original study and this replication attempt.

This direct replication provides an opportunity to understand the present evidence of these effects. Any known differences, including reagents and protocol differences, were identified prior to conducting the experimental work and described in the Registered Report (*Fiering et al., 2015*). However, this is limited to what was obtainable from the original paper and through communication with the original authors, which means there might be particular features of the original experimental protocol that could be critical, but unidentified. So while some aspects, such as cell line, mouse strain, antibodies, and the method to measure metastatic counts were maintained, others were changed during the execution of the replication that could affect results, such as the time from cell injection until euthanasia, which was shorter in this replication attempt than what was conducted in the original study. Additionally, other aspects were unknown or not easily controlled for. These include variables such as cell line genetic drift (*Ben-David et al., 2018*; *Hughes et al., 2007*; *Kleensang et al., 2016*), including subclonal drift in heterogeneous stable cells (*Shearer and Saunders, 2015*), genetic heterogeneity of mouse inbred strains (*Casellas, 2011*), the microbiome of recipient mice (*Macpherson and McCoy, 2015*), and housing temperature in mouse facilities (*Kokolus et al., 2013*). Mutations could have also accumulated during cell passage *in vitro* and drive cell lines towards a different phenotype that is observed *in vivo* (*Gregoire et al., 2001*; *Hurlin et al., 1991*). Environmental differences such as husbandry staff, bedding type and source, light levels, and other intangibles, all of which, by necessity, differed between the studies, which along with bias during welfare assessment and measurement imprecision can also affect experimental outcomes with mice (*Howard, 2002*; *Jensen and Ritskes-Hoitinga, 2007*; *Nevalainen, 2014*; *Sorge et al., 2014*). Differences in imaging instruments is another source of variability that could affect the outcomes between studies. The implementation of standardization procedures for equipment performance (e.g. International Organization for Standardization/Draft International Standard for confocal microscopes currently under development [ISO/DIS 21073]) could provide metrics to compare one instrument to another, facilitating reproducibility. Furthermore, differences in image analysis and batch processing could be another source of variability between studies, illustrating the benefit of documenting all analysis configuration parameters and keeping results connected to the input data (*Nanes, 2015*). Also, there is the possibility that human cancer cells, such as the LM-4175 cells used in the original study and this replication attempt, may behave differently in mouse models compared to other studies where mouse cancer cells were injected in mice (*Capozza et al., 2012*). Whether these or other factors influence the outcomes of this study is open to hypothesizing and further investigation, which is facilitated by direct replications and transparent reporting.

# Materials and methods

## Key resources table

| Reagent type (species) or resource | Designation | Source or reference | Identifiers | Additional information |
|---|---|---|---|---|
| Cell line (*Mus musculus*) | Cav1WT pMEFs | This paper | | isolated from embryonic day 14.5 embryos from B6129SF2/J mice (Jackson Laboratory, Stock No. 101045, RRID:IMSR_JAX:101045) |
| Cell line (*M. musculus*) | Cav1KO pMEFs | This paper | | isolated from embryonic day 14.5 embryos from Cav1$^{tm1Mls}$/J mice (Jackson Laboratory, Stock No. 004585, RRID:IMSR_JAX:004585) |
| Cell line (*H. sapiens*, female) | LM-4175 | doi: 10.1038/nature03799 | | Expresses HSV-tk1-GFP-Fluc; shared by del Pozo lab, CNIC |
| Strain, strain background (*M. musculus*, Athymic Nude-*Foxn1$^{nu}$*, female) | athymic nude | Envigo | MGI:5652489 | |
| Other | Matrigel | Corning | cat# 356234 | |
| Other | D-luciferin | Promega | cat# P1042 | |
| Antibody | mouse anti-Caveolin 1 | BD Biosciences | cat# 610406; clone: 2297; RRID:AB_397789 | 1:1000 dilution |
| Antibody | mouse anti-alpha-SMA | Sigma-Aldrich | cat# A5228; clone: 1A4; RRID:AB_262054 | 1:100 or 1:1000 dilution |
| Antibody | mouse anti-gamma-tubulin | Sigma-Aldrich | cat# T6557; clone: GTU-88; RRID:AB_477584 | 1:1000 dilution |
| Antibody | rabbit-fibronectin | Sigma-Aldrich | cat# F3648, RRID:AB_476976 | 1:200 dilution |
| Antibody | HRP-conjugated goat anti-mouse | Thermo Fisher Scientific | cat# 32430; RRID:AB_1185566 | 1:5000 to 1:10,000 dilution |
| Antibody | Alexa Fluor 594-conjugated donkey anti-rabbit | Jackson Immuno Research Laboratories | cat# 711-585-152; RRID:AB_2340621 | 1:300 dilution |
| Antibody | Alexa Fluor 647-conjugated donkey anti-mouse | Jackson Immuno Research Laboratories | cat# 715-605-151; RRID:AB_2340863 | 1:300 dilution |
| Antibody | rabbit IgG isotype control | Sigma-Aldrich | cat# I5006; RRID:AB_1163659 | 1:200 dilution |
| Antibody | mouse IgG2a isotype control | Sigma-Aldrich | cat# M5409; clone: UPC-10; RRID:AB_1163691 | 1:100 dilution |
| Software, algorithm | Qcapture-pro | Teledyne Qimaging | RRID:SCR_014432 | version 6.0.0.605 |
| Software, algorithm | Living Image | Perkin Elmer | RRID:SCR_014247 | version 4.3.1 |
| Software, algorithm | Zen Black Acquisition | ZEISS | RRID:SCR_013672 | version 2.0 |
| Software, algorithm | KNIME | KNIME | RRID:SCR_006164 | version 3.5.1 |
| Software, algorithm | ImageJ | doi:10.1038/nmeth.2089 | RRID:SCR_003070 | version 1.50a |
| Software, algorithm | Fiji | doi:10.1038/nmeth.2019 | RRID:SCR_002285 | version 2.0.0-rc-34 |
| Software, algorithm | FibrilTool | doi:10.1038/nprot.2014.024 | RRID:SCR_016773 | |
| Software, algorithm | OrientationJ | doi:10.1007/s10237-011-0325-z | RRID:SCR_014796 | version 2.0.3 |

*Continued on next page*

*Continued*

| Reagent type (species) or resource | Designation | Source or reference | Identifiers | Additional information |
|---|---|---|---|---|
| Software, algorithm | MetaMorph | Molecular Devices | RRID:SCR_002368 | version 7.10.1 |
| Software, algorithm | Bio-Formats Importer plugin | doi:10.1083/jcb.201004104 | RRID:SCR_000450 | version 5.1.9 |
| Software, algorithm | R Project for statistical computing | https://www.r-project.org | RRID:SCR_001905 | version 3.5.1 |

As described in the Registered Report (*Fiering et al., 2015*), we attempted a replication of the experiments reported in Figure 7C and Supplemental Figures S2A and S7C of *Goetz et al. (2011)*. A detailed description of all protocols can be found in the Registered Report (*Fiering et al., 2015*) and are described below with additional information not listed in the Registered Report, but needed during experimentation.

## Cell culture

Cav1WT and Cav1KO pMEFs were isolated from embryonic day 14.5 embryos from B6129SF2/J mice (Jackson Laboratory, Stock No. 101045, RRID:IMSR_JAX:101045) and Cav1$^{tm1Mls}$/J mice (Jackson Laboratory, Stock No. 004585, RRID:IMSR_JAX:004585), respectively, following the procedure outlined in the Registered Report (*Fiering et al., 2015*). Multiple pMEF clones were isolated and tested (*Figure 1*) and clone #6 for Cav1WT and clones #12 and #13 for Cav1KO were used for the animal study. pMEFs were used in all experiments before passage 5. LM-4175 cells (lung metastasis derived from MDA-MB-231 cells) retrovirally infected with a triple-fusion protein reporter construct encoding herpes simplex virus thymidine kinase 1, green fluorescent protein (GFP), and firefly luciferase (HSV-tk1-GFP-Fluc) (*Minn et al., 2005*) were shared by Dr. Miguel A. del Pozo, Centro Nacional de Investigaciones Cardiovasculares Carlos III (CNIC) at passage 20 and used at passage 23 for experiments. Cav1WT pMEFs, Cav1KO pMEFs, and LM-4175 cells were grown in DMEM (Thermo Fisher Scientific, cat# 11054001) supplemented with 10% fetal bovine serum (FBS), 4 mM L-glutamine, 100 U/ml penicillin and 100 µg/ml streptomycin at 37°C in a humidified atmosphere at 5% $CO_2$. Quality control data for the cell lines are available at https://osf.io/hkdwv/. This includes results confirming the cell lines are free of mycoplasma contamination and common mouse pathogens, as well as STR DNA profiling of the cell lines with LM-4175 matched to MDA-MB-231 (RRID:CVCL_0062) when queried against an STR profile database (IDEXX BioResearch, Columbia, Missouri).

## Western blots

Cav1WT and Cav1KO pMEFs (at passage 3) were prepared in RIPA lysis buffer (50 mM Tris-HCl, pH 8.0, 150 mM NaCl, 1% Triton X-100, 0.1% SDS, 0.5% Sodium deoxycholate, 1 mM NaF, and 1 mM $Na_3VO_4$), supplemented with protease (Roche, cat# 04693116001) and phosphatase inhibitors (Roche, cat# 04906845001) at manufacturer recommended concentrations. Lysed cells were scraped from plates and centrifuged at 14,000x*g* for 15 min at 4°C before protein concentration of supernatant was quantified using a Bradford assay following manufacturer's instructions. Lysate samples were separated by SDS-PAGE gel electrophoresis in 1X Tris-glycine SDS buffer run at 100V through the stacking part of the gel and 180V after the proteins had migrated through the resolving gel (15%) until the dye front was at the bottom of the gel, but had not migrated off. Gels were transferred to an Immobilon-P PVDF membrane (Millipore, cat# IPVH00010) and then incubated with 5% non-fat dry milk in 1X TBS with 0.1% Tween-20 (TBST). Membranes were probed with the following primary antibodies diluted in 5% non-fat dry milk in TBST: mouse anti-Caveolin 1 [clone 2297] (BD Biosciences, cat# 610406, RRID:AB_397789), 1:1000 dilution; mouse anti-alpha-SMA [clone 1A4] (Sigma-Aldrich, cat# A5228, RRID:AB_262054), 1:1000 dilution; mouse anti-gamma-tubulin [clone GTU-88] (Sigma-Aldrich, cat# T6557, RRID:AB_477584), 1:1000 dilution. Membranes were washed with TBST and incubated with secondary antibody diluted in 5% non-fat dry milk in TBST: HRP-conjugated goat anti-mouse (Thermo Fisher Scientific, cat# 32430, RRID:AB_1185566), 1:5000 to 1:10,000 dilution. Membranes were washed with TBST and incubated with ECL reagent (Santa Cruz Biotechnology, cat# sc-2048) to visualize signals. Scanned Western blots were quantified using ImageJ

software (RRID:SCR_003070), version 1.50a (*Schneider et al., 2012*). Additional methods and data, including full Western blot images, are available at https://osf.io/na5h2/.

## Collagen gel contraction assay

$1.5 \times 10^5$ Cav1WT or Cav1KO pMEFs were mixed with NaOH-titrated collagen I (Corning, cat# 354249) to a final collagen I concentration of 1 mg/ml in a total of 500 µl. The mixture was immediately transferred to a 24 well ultra low attachment plate (Corning, cat# 3473) and allowed to solidify at room temperature for about 1 hr. After solidification, 500 µl of cell growth medium was added to each well and gels were dissociated from the well by gently running a 200 µl pipet tip along the gel edge without shearing or tearing the gel. Plates were swirled to ensure the gel was free from the plate and then incubated at 37°C in a humidified atmosphere at 5% $CO_2$ for 48 hr. Images were taken at 24 hr and 48 hr to document contraction. Assay was performed in triplicate for each clone and no cell controls (cell growth medium only) were included. Gel contraction index was calculated from the gel surface area measured on acquired images using a digital camera (Leica MZ16 stereo-microscope and QCapture-pro software (Teledyne QImaging, RRID:SCR_014432), version 6.0.0.605) at a fixed distance above the gels, and reported as the percentage of contraction of the initial surface area. This experiment was pre-registered before experimental work began (https://osf.io/9cgk4/). Additional detailed methods and data, including images of gels, are available at https://osf.io/na5h2/.

## Subcutaneous tumorigenicity assay

All animal procedures were approved by the Dartmouth College IACUC# 1133 and were in accordance with the Dartmouth College policies on the care, welfare, and treatment of laboratory animals. Eight-ten-week old female athymic nude mice (Envigo, Strain: Hsd:Athymic Nude-*Foxn1^{nu}*, MGI:5652489) were housed (4-5 per cage) in standard ventilated filtered cages, with corn Cobb bedding and a nestlet for nesting (changed evey other week), 12 hr light/dark cycles, and fed sterile rodent chow (Teklad global 18% protein rodent diet (Envigo, cat# 2918)) and acidified water (changed weekly) *ad libitum*. The mice were housed for approximately 2 weeks before being enrolled in the study. The individual mouse was considered the experimental unit within the studies and inclusion/exclusion criteria (e.g. mice were excluded if injection of tumor cells entered the peritoneum) are described in the Registered Report (*Fiering et al., 2015*). Housing and experimentation (e.g. injection, IVIS imaging, etc) were conducted in the same facility, which was kept at 72°F +/- 2°F with 30-70% relative humidity.

A pilot study was performed on five mice. Mice were anesthetized with 2-2.5% isoflurane (Patterson Veterinary, cat# 07-893-1389) mixed with 1L/min of medical grade oxygen and injected subcutaneously with $1 \times 10^6$ LM-4175 cells unmixed (four mice) or mixed 1:1 with Cav1WT pMEFs (one mouse) in 100 µl PBS mixed with 100 µl of Matrigel (Corning, cat# 356234) in the flank using a 25-gauge needle. Mice were monitored until visible tumors formed. Once tumor growth was detected in any animal, tumors were measured using precision calipers twice a week, and mice were monitored for signs of distress daily. Mice were euthanized starting at day 40 post-injection due to ascites thru day 63 due to excessive tumor burden. After reviewing tumor measurements, it was determined that day 45 was the target end date according to the IACUC approved protocol.

Following the pilot study, a total of 62 mice were randomized (simple randomization using a random number generator) to receive a subcutaneous injection with $1 \times 10^6$ LM-4175 cells unmixed, the control group, (10 mice) or mixed 1:1 with Cav1WT pMEFs (26 mice) or Cav1KO pMEFs (26 mice) as described in the pilot study. Injections occurred on two separate days, with half the mice for each group injected on each day. Mice were euthanized at day 45 post-injection to ensure consistency of collected data and minimize animal suffering per IACUC guidelines. Of note, one mouse, LM-4175 plus Cav1KO pMEFs, was euthanized, and thus excluded, because the tumor was above the approved IACUC protocol tumor limit before the specified endpoint of 45 days. No other mice were excluded, although similar to the pilot study we observed adverse events (e.g. ulceration at the tumor site) confirming that 45 days after injection was the appropriate endpoint. To measure primary tumor growth and metastasis, mice were anesthetized and injected with 100 µl of 30 mg/ml D-luciferin (Promega, cat# P1042) intraperitoneally. Twenty minutes later, mice were placed into an IVIS Spectrum system (Caliper, Xenogen) for imaging of ventral views for photon flux quantification.

Following imaging mice were injected with 50 µl of 30 mg/ml D-luciferin intraperitoneally. Twenty minutes later, mice were euthanized and the primary tumor and following organs were dissected: lymph nodes, spleen, lungs, liver, intestines, kidneys. Organs were placed separated, into a 100 mm dish and placed into IVIS for imaging. The primary tumors were cut in half and frozen in O.C.T. compound by placing cassette with tumor into a dry ice/ethanol bath until frozen and then storing at −80℃ until shipped on dry ice for image processing. Anesthesia, luciferin injections, imaging, second luciferin injections, euthanasia, dissection, and imaging/freezing primary tumors were performed during daylight (afternoon hours) with mice from different groups in parallel so variations during the procedure were equal across groups.

## IVIS imaging

Images were acquired with a Xenogen IVIS Imaging System (Perkin Elmer, 200 Series) and Living Image software (RRID:SCR_014247), version 4.3.1 at a medium binning level and the field of view set at 'E'. For *in vivo* imaging, mice were placed into the IVIS with front limbs taped above head and black shields used to block bioluminescence from the primary tumors to visualize metastases *in vivo*. Exposure time for photon flux quantification of primary tumors, a primary outcome measure, was 0.2 s. After *in vivo* imaging, dissected organs were imaged *ex vivo* to detect metastatic foci, a primary outcome measure. Images were taken at multiple exposures (0.2 s, 0.5 s, 1 s, 10 s, 20 s, 60 s, and 120 s) and used to manually quantify visible metastatic foci. Quantification was performed blinded to the cells the animals were injected with. Image files are available at https://osf.io/bq54u/.

## Immunofluorescence and confocal microscopy

A random subset (simple randomization from each group) of the cryopreserved primary tumors from the subcutaneous tumorigenicity assay were sectioned (8 µm thick), fixed, permeabilized, and stained as described in the Registered Report (*Fiering et al., 2015*) with the following primary antibodies diluted in PBS supplemented with 2% BSA overnight at 4℃: rabbit anti-fibronectin (Sigma-Aldrich, cat# F3648, RRID:AB_476976), 1:200 dilution; mouse anti-alpha-SMA [clone 1A4] (Sigma-Aldrich, cat# A5228, RRID:AB_262054), 1:100 dilution. Sections were washed in PBS and incubated with the following secondary antibodies diluted in PBS supplemented with 2% BSA for 1 hr at 37℃: Alexa Fluor 594 conjugated donkey anti-rabbit (Jackson ImmunoResearch Laboratories, cat# 711-585-152, RRID:AB_2340621), 1:300 dilution; Alexa Fluor 647 conjugated donkey anti-mouse (Jackson ImmunoResearch Laboratories, cat# 715-605-151, RRID:AB_2340863), 1:300 dilution. Hoechst dye (1:5000 dilution) was used to counterstain nuclei. Additional controls were included on a subset of the primary tumor sections: rabbit IgG isotype control (Sigma-Aldrich, cat# I5006, RRID:AB_1163659), 1:200; mouse IgG2a isotype control [clone UPC-10] (Sigma-Aldrich, cat# M5409, RRID: AB_1163691), 1:100; secondary antibody only controls. Additionally, in an attempt to reduce the non-specific staining observed with the mouse anti-SMA antibody, we included a mouse-on-mouse blocking step (Vector lab, cat# MKB-2213) before incubation with the primary antibodies as a test on subset of the samples (*Figure 3—figure supplement 1C*). Samples were imaged using a LSM 880 upright confocal microscope (ZEISS, Oberkochen, Germany) fitted with a 40X Plan Apochromat NA 1.3 oil immersion objective. Ten random (simple randomization) z-stacks, with a total of 26 slices at 0.3 µm intervals per z-stack, were acquired per sample. Image acquisition was performed with a laser-scanning confocal laser running with Zen Black Acquisition Software (RRID:SCR_013672), version 2.0. Detailed image acquisition settings are contained in the metadata of the raw images. Image acquisition was performed blinded to the sample identity and the different fields of view were chosen randomly. Image files are available at https://osf.io/bq54u/.

## Fibronectin fiber analysis

All image analysis was performed blinded to the sample identity. Image analysis output files are available at https://osf.io/bq54u/. Images were processed using KNIME (www.KNIME.com; RRID: SCR_006164), version 3.5.1 (*Berthold et al., 2007*). A screenshot of the processing/analysis steps are illustrated in *Figure 3—figure supplement 4*. Briefly, the fibronectin channel was selected and Z-stacks were processed for maximum intensity projections. The ImageJ (RRID:SCR_003070) command 'Subtract background', with a rolling setting of 50 (value that was optimized for the current dataset) was applied. Images were then thresholded for 35% intensity, outputting the binary images

necessary for subsequent measurements (*Figure 3—figure supplement 3B*). The ImageJ command 'Analyze particles' was then applied, with options set to 'Iterations = 1', 'Count = 1' black', 'Set measurements: area, mean, fit, redirect = None, decimal = 3', 'Analyze particles: size = 25.0 infinity, circularity = 0.0–1.0, show=(Outlines)". This was repeated for objects larger than 25, 50, 150, 300, and 357 pixels and the angle measurements of each object were exported for further analysis. To determine percent of fibers within 20° of the modal angle, the primary outcome measure, the relative angles were rounded to the nearest 10° angle using the rounding base function of R and then determining the mode angle for each image (i.e. the angle with the most fibers observed) as described previously (*Amatangelo et al., 2005*; *Fiering et al., 2015*). Script used to determine the mode angle for each image is available at https://osf.io/qgjme/.

Additionally, for each image, an anisotropy factor and the average angle of the fibers was measured by implementing the FibrilTool macro (RRID:SCR_016773) (*Boudaoud et al., 2014*) (FibrilTool was converted to a KNIME-node, see: https://osf.io/au4dx/), coherency was measured with the OrientationJ plugin (RRID:SCR_014796), version 2.0.3 (*Rezakhaniha et al., 2012*), and blinded manual scoring to assess 'the frequency of parallel fibers' used the following scale: (1) Not at all (in about 0%), (2) Occasionally (in about 30%), (3) Sometimes (in about 50%), (4) Usually (in about 80%), (5) All are (in about 100%). The values from the ten images for each tumor were averaged to generate a single score for each tumor.

An attempt to determine fibronectin orientation was made using MetaMorph (Molecular Devices, RRID:SCR_002368), version 7.10.1. Images were processed one at a time, not batch processed. In Fiji (RRID:SCR_002285) (*Schindelin et al., 2012*)/ImageJ, version 2.0.0-rc-34/1.50a (build 927ecc3c7a), the Bio-Formats Importer plugin (RRID:SCR_000450) (*Linkert et al., 2010*), version 5.1.9 was used to read/open the confocal microscopy raw data. The plugin was configured to split the channels to pursue the processing only on fibronectin. The resulting fibronectin Z-stacks were saved and then read/opened in MetaMorph, and subjected to a maximum intensity projection to generate a single 2D image (Note: this was our interpretation of the original description 'Overlay Z-slices to make reconstituted views of the corresponding 3-D fibers for each region'). The MetaMorph Background and shading correction function was executed with a setting of 15 pixels (Note: this was our interpretation of the original description 'Reduce non-specific background by selectively darkening objects with a pixel area greater than 15 using the flatten background function'). Using the MetaMorph internal threshold function, a binary image was created at the 35% setting (i.e. 35% of the pixels have the intensity). The resulting binary image was subjected to the MetaMorph IMA (Integrated Morphometry Analysis) function to reveal objects of interest, which was unable to be executed as there were too many objects to process.

## Statistical analysis

Statistical analysis was performed with R software (RRID:SCR_001905), version 3.5.1 (*R Development Core Team, 2018*). All data, csv files, and analysis scripts are available on the OSF (https://osf.io/7yqmp/). Confirmatory statistical analysis was pre-registered (https://osf.io/s6ndp/) before the experimental work began as outlined in the Registered Report (*Fiering et al., 2015*). Data were checked to ensure assumptions of statistical tests were met. When described in the results, the Bonferroni correction, to account for multiple testings, was applied to the alpha error or the *p*-value. The Bonferroni corrected value was determined by divided the uncorrected value (0.05) by the number of tests performed. A meta-analysis of a common original and replication effect size was performed with a random effects model and the *metafor* R package (*Viechtbauer, 2010*) (https://osf.io/rvf57/). Meta-analyses were performed without weighting for Cliff's *d*, since unweighted Cliff's *d* has been reported to reduce bias (*Kromrey et al., 2005*). The asymmetric confidence intervals for the overall Cliff's *d* estimate was determined using the normal deviate corresponding to the $(1 - \text{alpha}/2)^{\text{th}}$ percentile of the normal distribution (*Cliff, 1993*). The raw data pertaining to Figure 7Cb, 7 Cd, S7Ca, and S7Cc of *Goetz et al. (2011)* were shared by the original authors and compared back to the published summary data and figures. The summary data was published in the Registered Report (*Fiering et al., 2015*) and used in the power calculations to determine the sample sizes for this study.

## Data availability

Additional detailed experimental notes, data, and analysis are available on OSF (RRID:SCR_003238) (https://osf.io/7yqmp/; *Sheen et al., 2018*). This includes the R Markdown file (https://osf.io/rd3yf/) that was used to compose this manuscript, which is a reproducible document linking the results in the article directly to the data and code that produced them (*Hartgerink, 2017*). The image analysis workflow generated during this study is available on Amazon Web Services (AWS) as an Amazon Machine Image (AMI). The machine image is located in the N. Virginia (us-east-1) region with the AMI ID: ami-09ee55780b0c19120, and AMI Name: rpcb-analysis-study20. Computation was performed on an Instance Type of m5.4xlarge (16 vCPU, 64 GiB Memory), with 500 GiB of Elastic Black Store (EBS) storage, and running Windows Server 2016. The administrator account password required to login is 'RPCB!Analysis'.

## Deviations from registered report

Following completion of the Western blot analysis to assess SMA levels in Cav1WT and Cav1KO pMEF clones, we consulted with the original authors regarding the lack of observable change between the two types of pMEFs. As suggested by the original authors we conducted a collagen gel contraction assay to assess ECM remodeling capabilities, which was pre-registered before experimental work began (https://osf.io/9cgk4/). For the subcutaneous tumorigenicity assay, the planned study design indicated the mice would be euthanized 70 days after cell injection, or an earlier time point to not compromise the ability to obtain enough mice for analysis while ensuring no animal suffering. Following a pilot study this was determined to be 45 days after injection, which was confirmed in the experimental study. A different anesthesia than listed in the Registered Report was used during cell and luciferin injections (isoflurane instead of ketamine and xylazine due to availability) as well as a different dose of luciferin (100 µl of 30 mg/ml for the first injection instead of 150 µl of 17.5 mg/ml and 50 µl of 30 mg/ml for the second injection instead of 50 µl of 17.5 mg/ml). Also, as described above (*Figure 3—figure supplement 1*), we were unable to obtain SMA staining that was specific, based on controls, to allow for the quantification as specified in the Registered Report to be conducted. As such, we did not conduct the analysis that was dependent on the SMA staining outlined in the Registered Report. As described above, we attempted to determine fibronectin orientation using MetaMorph software, but found the processing could not be executed as previously described as there were too many objects to process. Instead, we created a workflow using the KNIME analytics platform. We also explored additional methods to examine fiber orientation as described above. Additional materials and instrumentation not listed in the Registered Report, but needed during experimentation are also listed.

## Acknowledgements

The Reproducibility Project: Cancer Biology would like to thank Dr. Miguel A del Pozo (Centro Nacional de Investigaciones Cardiovasculares Carlos III (CNIC), Madrid, Spain), for sharing critical information, data, and reagents, specifically the LM-4175 cells expressing HSV-tk1-GFP-Fluc. We want to thank Claire Brown (Director of the Advanced Bio-Imaging Facility, McGill University, Montréal, Canada) and Vincent Pelletier (Implementation Specialist, Quorum Technologies, Puslinch, Canada) for sharing their expertise with the MetaMorph image analysis software. We thank Stefan Helfrich (Academic Alliance Manager, KNIME GmbH, Konstanz, Germany) for assistance with KNIME Analytics Platform. This paper was included in the 'Road testing for ARRIVE 2019', which helped improve our reporting of the animal experimental details. We would also like to thank the following companies for generously donating reagents to the Reproducibility Project: Cancer Biology; American Type and Tissue Collection (ATCC), Applied Biological Materials, BioLegend, Charles River Laboratories, Corning Incorporated, DDC Medical, EMD Millipore, Harlan Laboratories, LI-COR Biosciences, Mirus Bio, Novus Biologicals, Sigma-Aldrich, and System Biosciences (SBI).

## Additional information

### Group author details

**Reproducibility Project: Cancer Biology**
**Elizabeth Iorns**: Science Exchange, Palo Alto, United States; **Rachel Tsui**: Science Exchange, Palo Alto, United States; **Alexandria Denis**: Center for Open Science, Charlottesville, United States; **Nicole Perfito**: Science Exchange, Palo Alto, United States; **Timothy M Errington**: Center for Open Science, Charlottesville, United States

### Competing interests

Mee Rie Sheen, Jennifer L Fields, Steven Fiering: Transgenics and Genetic Constructs Shared Resource Center, Geisel School of Medicine at Dartmouth is a Science Exchange associated lab. Brian Northan, Judith Lacoste: Cellavie Inc is a Science Exchange associated lab. Lay-Hong Ang: Confocal Imaging Core Facility, Beth Israel Deaconess Medical Center was a Science Exchange associated lab. Reproducibility Project: Cancer Biology: EI, RT, NP: Employed by and hold shares in Science Exchange Inc.The other authors declare that no competing interests exist.

### Funding

| Funder | Author |
| --- | --- |
| Laura and John Arnold Foundation | Reproducibility Project: Cancer Biology |

The funder had no role in study design, data collection and interpretation, or the decision to submit the work for publication.

### Author contributions

Mee Rie Sheen, Acquisition of data, Analysis and interpretation of data, Drafting or revising the article, Performed isolation and characterization of pMEFs; Jennifer L Fields, Acquisition of data, Analysis and interpretation of data, Drafting or revising the article, Performed subcutaneous tumorigenicity assay; Brian Northan, Judith Lacoste, Reproducibility Project: Cancer Biology, Acquisition of data, Analysis and interpretation of data, Drafting or revising the article, Performed image analysis; Lay-Hong Ang, Acquisition of data, Analysis and interpretation of data, Drafting or revising the article, Performed staining and imaging of tumors; Steven Fiering, Analysis and interpretation of data, Drafting or revising the article

### Author ORCIDs

Alexandria Denis (iD) http://orcid.org/0000-0002-1210-2309
Timothy M Errington (iD) http://orcid.org/0000-0002-4959-5143

### Ethics

Animal experimentation: All animal procedures were approved by the Dartmouth College IACUC# 1133 and were in accordance with the Dartmouth College policies on the care, welfare, and treatment of laboratory animals.

### Decision letter and Author response

Decision letter https://doi.org/10.7554/eLife.45120.sa1
Author response https://doi.org/10.7554/eLife.45120.sa2

## Additional files

### Supplementary files

- Transparent reporting form
- Reporting standard 1. The ARRIVE guidelines checklist.

## Data availability

Additional detailed experimental notes, data, and analysis are available on OSF (RRID:SCR_003238) (https://osf.io/7yqmp/; Sheen et al., 2018). This includes the R Markdown file (https://osf.io/rd3yf/) that was used to compose this manuscript, which is a reproducible document linking the results in the article directly to the data and code that produced them (Hartgerink, 2017). The image analysis workflow generated during this study is available on Amazon Web Services (AWS) as an Amazon Machine Image (AMI). The machine image is located in the N. Virginia (us-east-1) region with the AMI ID: ami-09ee55780b0c19120, and AMI Name: rpcb-analysis-study20. Computation was performed on an Instance Type of m5.4xlarge (16 vCPU, 64 GiB Memory), with 500 GiB of Elastic Black Store (EBS) storage, and running Windows Server 2016. The administrator account password required to login is "RPCB!Analysis".

The following dataset was generated:

| Author(s) | Year | Dataset title | Dataset URL | Database and Identifier |
|---|---|---|---|---|
| Sheen MR, Fields JL, Northan B, Lacoste J, Ang L-H, Fiering SN, Iorns E, Tsui R, Denis A, Haselton M, Perfito N, Errington TM | 2018 | Study 20: Replication of Goetz et al., 2011 (Cell) | http://dx.doi.org/10.17605/OSF.IO/7YQMP | Open Science Framework, 10.17605/OSF.IO/7YQMP |

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
