## [Decision Letter]

Thank you for submitting your article "Replication Study: Biomechanical remodeling of the microenvironment by stromal caveolin-1 favors tumor invasion and metastasis" for consideration by *eLife*. Your article has been reviewed by Sean Morrison as the Senior Editor, a Reviewing Editor, and four reviewers. The following individuals involved in review of your submission have agreed to reveal their identity: Miguel Del Pozo (Reviewer #1); Kenneth Yamada (Reviewer #4).

The reviewers have discussed the reviews with one another and the Reviewing Editor has drafted this decision to help you prepare a revised submission.

Summary:

This Replication Study has reproduced some parts of the original paper, but for other parts no statements about reproducibility can be drawn, since the experimental setup was different from the original study.

Essential revisions:

1) The original authors carried out in vivo experiments for 70 days, but this replication study terminated them at 45 days. While the rationale for shortening experiments by 25 days (a net reduction of 36% from original conditions!) is respectable, this makes it impossible to draw any firm conclusion with respect to the reproducibility of the original claims related to metastasis, since metastasis is a very time-dependent process. You need to more clearly acknowledge that you cut the metastasis assay short relative to the original study and therefore cannot draw a conclusion concerning the reproducibility of the metastasis results. This should be done in the Abstract.

2) You DID find similar SMA expression levels across genotypes when cultured in 2D plastic. The description of these experiments (subsection “Isolation and characterization of Cav1 wild-type and Cav1 knockout primary MEFs”) is misleading. The observations were not unexpected: the original study had already compared SMA expression between WT and Cav1KO MEFs in 2D and 3D conditions (Figure S2A, original study).

3) The data in Figure 3 exhibit poor staining quality and an overcompensation in digital levels. Hence, the quantitation is not comparable to that of the original study, and this should clearly be noted in the manuscript.

4) Clearly discuss the problem with the primary fibroblasts used in this study, which appear to be less contractile (see Figure 1C-E) than in the initial study (see Figure 1F). The issue is that these fibroblasts could not be further characterized because of antibody cross reactions during SMA staining (Figure 3). In the absence of further characterization of the fibroblasts used in this study, the authors conclusions are undermined.

5) Acknowledge and discuss the fact that not using the specific steps of MetaMorph analysis to score fibronectin fiber orientation could have altered the results.

*Reviewer #1:*

The goal of this Reproducibility Project is undermined because differential results are derived from experiments failing to reproduce key aspects of the original study, thus unsuitable for fair comparison. Furthermore, a bias is put into highlighting such 'differences', belittling observations faithfully replicating conditions and yielding results similar to those of the original study (SMA expression levels; collagen contraction assays; even trends for lower ECM fiber order in tumors bearing Cav1KO fibroblasts despite substantial shortcomings).

Goetz et al., carried out in vivo experiments for 70 days, but this replication study terminated them at 45 days. While the rationale for shortening experiments by 25 days (a net reduction of 36% from original conditions!) is respectable, this fully invalidates this section to provide a fair assessment of reproducibility of the original paper. Parameters hinting at how limited comparison is include different organ distribution of metastasis, and the failure to detect their luminescence in vivo. Such a key difference as experiment duration overrules other argued potential sources of variation (see subsection “Subcutaneous tumorigenicity assay of tumor cells co-injected with Cav1WT or Cav1KO primary MEFs"). Metastasis is a non-linear event, and conclusions cannot be drawn if timespan allowed for growth and progression is drastically different.

Therefore, the authors should acknowledge that in vivo experiments simply cannot be compared. The authors briefly mention these pivotal pitfalls somewhere in the text themselves (see for example subsection “Meta-analyses of original and replication effects” and subsection “Subcutaneous tumorigenicity assay of tumor cells co-injected with Cav1WT or Cav1KO primary MEFs"), but nonetheless highlight such differential observations (even in the Abstract!) as relevant, and compare their potential interpretation. It should be explicitly stated in the Abstract, and clarified in the main text, that these sections are not valid attempts to reproduce the original report. I would even suggest moving upfront passages such as those listed above (subsection “Meta-analyses of original and replication effects” and subsection “Subcutaneous tumorigenicity assay of tumor cells co-injected with Cav1WT or Cav1KO primary MEFs"), as they address the core aim of the replication study: to faithfully reproduce original experimental conditions.

Other sections of the reproducibility study that are misleading, or for which clear limitations should be explicitly acknowledged, are the following:

First: the authors DID recapitulate similar SMA expression levels between genotypes when cultured in 2D plastic. The description of these incomplete experiments (subsection “Isolation and characterization of Cav1 wild-type and Cav1 knockout primary MEFs”) is however inaccurate and misleading. Their observations were NOT unexpected: the original study had already compared SMA expression between WT and Cav1KO MEFs in 2D and 3D conditions (Figure S2A, original study). The replication study would have benefited from performing CDM-based experiments-which is how Goetz et al., uncovered the reported changes-in addition to collagen contraction assays.

Second, experiments in Figure 3 exhibit poor staining quality and an over compensation in digital levels. Hence, their quantitation is simply not comparable to that of the original study (where additional complementary techniques were used, such as SHG microscopy). Setbacks when trying to use the original image analysis tools (based on MetaMorph software) justify their using a completely different image setting and analysis approach. While similar trends to the observations from Goetz et al., were recorded, these facts (which add up to the dismal difference in the duration of the experiments) again preclude a fair comparison between both studies.

In summary, this report unintentionally distorts the purpose of the Reproducibility Project itself, because its experimental execution, interpretation and writing-up are biased, emphasizing differential results despite being irrelevant and not comparable. A major revision of the text should therefore be carried out at the least.

*Reviewer #2:*

In this replication study, the authors have attempted to validate some of the original findings published in 2011 by the Del Pozo laboratory (Goetz et al., 2011). Based on previous exchange with the reviewers on one hand, and the authors of the original study on the other hand, it was agreed that two experiments of the original study (mainly Figure 7) were worth replicating: it involved mainly a subcutaneous xenograft of breast cancer derived lung metastatic cells in mice where tumor growth and metastasis would be analyzed.

In present study, some of the findings that were originally published by Goetz et al., could be reproduced. First, they could confirm that primary mouse embryonic fibroblasts (pMEF) that express Cav1 have increased extracellular matrix remodeling capability both in vivo and in vitro. They also found in agreement with the initial study that primary growth of the tumor was not affected by the presence of either Cav1 WT or Cav1 KO pMEFs. Unfortunately, they could not reproduce a key result of the original study that is that metastatic dissemination was increased by fibroblasts expressing Cav1 WT. Indeed, no difference could be found here between Cav1 WT and CAV1 KO pMEFs.

This is the most annoying result of this replication study as the controversy that exists about Cav1 and fibroblasts deals mainly with the role of Cav1 in metastasis. From the present study, it appears however that the exact experimental conditions of replication could not be respected, and that is certainly the main explanation for this discrepancy. I see two main problems with the replicated experiments. First, the primary fibroblasts used in this study appear to be less contractile (see Figure 1C-E) than in the initial study (see Figure 1F). This is certainly an important issue as the study examines the effect of these fibroblasts on tumor growth and metastasis. I am not convinced that the difference that is shown on Figure 1 C-E is significant. Another issue is that these fibroblasts could not be further characterized because of antibody cross reactions on SMA staining (Figure 3). I was anticipating these problems and this is why I had recommended in my initial review to also examine the involvement of p190RhoGAP in this process. For me in the absence of further characterization of the fibroblasts used in this study, the authors cannot conclude.

The most important issue with this study is the considerable difference in the time of observation of metastatic foci in various organs. While the original study observed metastasis after 70 days, here the observation is done at a considerably earlier time i.e. 45 days that is almost half the time. Considering that metastasis is a secondary event that takes time to occur, it is likely that the observation has been done too prematurely.

In conclusion, the replication study shows several similar aspects with the original study by Goetz et al., regarding the role of Cav1 WT expressing cancer associated fibroblasts in remodeling the extracellular matrix although it could not be statistically validated. The lack of reproducibility for the metastasis process is most likely due to the impossibility to respect the same experimental protocol than the one in Goetz et al. Combined with the fact that the fibroblasts injected here present a weaker contractile activity, and thus may also explain the absence of effect in metastasis. This is clearly an experimental weakness of this study.

In the Discussion section, the authors should discuss more these two important discrepancies rather than involving a list of experimental features (subsection “Meta-analyses of original and replication effects”), which if it is true that they may be involved would then be always present in all studies and as a matter of fact would prevent from reproducing any experiment. Also, the possibility that human cancer cells (the ones injected here) may behave differently in a mouse model should be discussed as it could explain the discrepancies observed with other studies where mouse cancer cells were injected in mice (see Capozza reference cited in the original study and not cited here).

*Reviewer #3:*

I have attempted to review this from the perspective of the statistical analysis and presentation of the data.

This paper should follow the registered report by Fiering et al., 2015.

I have found the report difficult to follow as the methods employed, results and discussion are all presented in different places.

From what I can judge the authors have presented their results in a clear manner and appear to have carried out the analysis consistently with how it was proposed in the registered report.

*Reviewer #4:*

The goal of this manuscript to replicate a cancer biology study published in Cell appears well-intentioned and potentially valuable. Unfortunately, the authors make two major changes to the original study protocol that have unknown and potentially substantial consequences, making it impossible in my opinion to provide a valid replication test of the original study – even though multiple other aspects of the approach such as pre-registration and the careful attempts to identify specific reagents and materials were commendable.

1) The first major change, even though seemingly within the parameters of the loosely written original plan that allowed for adjusting study length, has potential major problems. In any spontaneous metastasis assay, extending the time for analysis by 55% (25 days beyond 45 days to reach the 70 days in the original paper) could have major effects on both the extent and patterns of metastases. The current authors used a markedly truncated experimental endpoint for all animals even if many may well have not been in distress – the result is that comparisons of results between the two studies performed at 45 days versus 70 days seems impossible. Effects seen in the original study may have been missed or may not, which is impossible to determine. In fact, if the original Registered Report had explicitly proposed substituting an experimental endpoint for spontaneous metastasis of 45 days in place of the original 70 days, it seems unlikely that it would have been approved.

2) This major change in experimental endpoint timing also likely compromised the comparisons of fibronectin fiber patterns. Although it is impossible to know, it is plausible that extending the time of tumor interactions with the local microenvironment by 55% might affect the fibronectin pattern associated with intratumoral remodeling. Unless the original conditions could be met at least approximately, it seems like comparing apples and oranges (or green vs. over-ripe apples).

3) A second potentially important change was complete substitution of alternative image analysis methods concerning fibronectin patterns compared to the Registered Report describing the use of MetaMorph and scoring SMA-positive cells. It is simply not clear whether scoring all cells rather than SMA-positive cells could have affected the results. More importantly, not using the specific steps of MetaMorph analysis to score fibronectin fiber orientation could have altered the results. The authors provide somewhat plausible reasons for not scoring SMA-positive cells (the staining did not work in their hands) and avoiding the use of MetaMorph (they found "too many objects to process") but changing the workflow completely appears to invalidate the attempted replication. Unless the current authors can demonstrate that their altered methods provide the same outputs on the original data as the original methods [e.g., by obtaining primary data from the original authors], these substitutions involving altering the cell population analyzed (all cells vs. SMA+) and image analysis (KNIME analytics vs. MetaMorph) fail to provide direct experimental replication. In fact, a major specific defect of the new analysis is apparent from Figure 3—figure supplement 3 in which the software fails to capture the fibrillar nature of fibronectin staining in 4 out of 9 samples in which average brightness is merely lower: the authors show dots and short lines rather than the interconnected fibronectin fibrillar staining patterns.

4) Although not the approach taking by these well-intentioned replicators, this reviewer believes that more direct exchanges of expertise and experience between the original research group and the replication group could have avoided the serious problems listed above that, at present, unfortunately invalidate this well-intentioned replication attempt.

[Editors' note: further revisions were requested prior to acceptance, as described below.]

Thank you for resubmitting your work entitled "Replication Study: Biomechanical remodeling of the microenvironment by stromal caveolin-1 favors tumor invasion and metastasis" for further consideration at *eLife*. Your revised article has been favorably evaluated by Sean Morrison (Senior Editor), a Reviewing Editor, and three reviewers.

The manuscript has been improved but there are some remaining issues that need to be addressed before acceptance. The strong consensus among the referees and reviewing editor is that the shortening of the metastasis period makes it difficult to compare this study with the original one, and the manner in which this point is presented in the revised manuscript is not satisfactory. In particular, you must include additional discussion on your understanding of the relevance of a shorter or longer followup period for metastasis outcome, and the manner in which variations in this important parameter may influence the outcome of the experiments and, with that, the robustness of the conclusions that can be drawn in your study about the reproducibility of the original results.

*Reviewer #1:*

Fiering et al., complied in this revised version with several previous recommendations concerning the summary text, as well as changes to the main text body of the manuscript. However, the rewritting of the Abstract, while in the requested direction, should still be improved. We cannot help to notice the authors keep explicitly stating and ´presenting´ certain divergent observations as results worth being considered:

"[…] We found metastatic burden was similar between Cav1WT and Cav1KO pMEFs, while the original study found increased metastases with Cav1WT (Figure 7C; Goetz et al., 2011). We also found a statistically significant negative correlation of intratumoral remodeling with metastatic burden, while the original study found a statistically significant positive correlation (Figure 7CD; Goetz et al., 2011). Finally, we report meta-analyses for each result."

As they subsequently state, these differences can be explained alone by the fact that key experimental details were not reproduced (i.e. the replication study shortened the duration of in vivo experiments by more than 35% as compared to the original study, and they used a different image approach when assessing ECM architecture from microscopy images (apart from additional technical issues, already mentioned in the previous round of review)). Thus, we claim again that these parts of their replication study cannot be subject to fair comparison as such and should not be presented upfront with the same value as the rest of the observations. Moreover, the fact that key experimental aspects were not fully reproduced should be clearly stated before those descriptions. To provide an example of what, in our opinion, should be stated in the Abstract regarding these experiments:

"[…] Two key experimental parameters (experimental timing of in vivo metastasis experiments and image-based analysis of tumoral ECM architecture) did not match with those used in the original study, rendering some observations, such as metastatic burden or its correlation with intratumoral ECM remodeling, not suitable for comparison. Finally, we report meta-analyses for each result."

While we appreciate that an effort has been made trying to correct these issues according to the previous revision round, we think these details are paramount to avoid any ambiguous or misleading messages in the Abstract to potential readers. Besides, this was a common claim of reviewers 2, 4 and us (while the only other referee focuses on the statistical analysis and presentation of the data).

*Reviewer #5:*

This replication study addressed experiments performed by Goetz et al., (2011) which indicated a critical role of caveolin-1 expressed by myofibroblasts of the tumor stroma in ECM remodeling and its role in inducing distant metastasis formation. The results confirm the originally shown role of caveolin-1 in collagen contraction and, to some extent, in fibronectin matrix alignment in vivo, as well as a lack of involvement in tumor growth at the implantation site. Conversely, this replication study could replicate the effect of caveolin-1 positive fibroblasts on metastatic evasion, and even indicated an inverse effect between metastasis outcome and fibronectin alignment in the primary tumor.

With the notable exception of a reduced followup period of metastasis after tumor implantation. The experiments were designed and performed with high fidelity and the quality of documentation is excellent. The meta-analysis further explores similarities and differences with competence and is delineated in a comprehensive manner. As major shortcoming, the explanation justifying the shortened observation period for analyzing metastasis outcome and the discussion on the implications for similar replication work in general are not yet satisfactory.

Specific points:

Subsection “Subcutaneous tumorigenicity assay of tumor cells co-injected with Cav1WT or Cav1KO primary MEFs": "45 days after cell injection to maximize the length of time for tumor growth while minimizing animal suffering" – it should be stated whether in this replication study, the tumors grew more rapidly, compared to Goetz et al., and whether a different tumor size until human endpoint mandated this deviation compared to the original study.

For a general audience, the authors should include additional discussion on their understanding of the relevance of a shorter or longer followup period for metastasis outcome. Particularly, they should discuss how robust results can be considered and how similar replication work in independent labs should be performed. Is it legitimate to adjust followup periods? Which rules should apply to a robust biological outcome? How can the inverse correlation of metastasis and ECM remodeling in the primary tumor site be explained when compared to the primary work? Which growth monitoring criteria should be applied to achieve high fidelity replication? How should researchers deal with differences in human endpoint criteria when constrained by a given legal framework? How would the authors perform this part of the work in retrospect, to achieve higher concordance with the original study?

The discussion on intermittent parameters potentially affecting the bioluminescence imaging is of general interest. It is not clear, however, whether these points are relevant here, assuming that the same imaging approach was used as in Goetz et al. (whole-body bioluminescence). However, Goetz et al., additionally used bioluminescence analysis of excised organs. Can the authors clarify why this more sensitive approach was apparently not used here?

[Editors' note: further revisions were requested prior to acceptance, as described below.]

Thank you for resubmitting your work entitled "Replication Study: Biomechanical remodeling of the microenvironment by stromal caveolin-1 favors tumor invasion and metastasis" for further consideration at *eLife*. Your revised article has been favorably evaluated by Sean Morrison (Senior Editor), a Reviewing Editor, and two reviewers.

The manuscript has been improved but there are a few remaining issues that need to be addressed before acceptance, as outlined below:

1) The most critical issue is that the editors and reviewers were not satisfied with the clarity of the Abstract in terms of the way it acknowledges the potential impact of the shorter metastasis assay on the interpretability of the results. Based on extensive consultation among the editors and reviewers, their concerns would be resolved if you would be willing to replace the last four sentences in the Abstract with the following text:

We found metastatic burden was similar between Cav1WT and Cav1KO pMEFs, while the original study found increased metastases with Cav1WT (Figure 7C; Goetz et al., 2011); however, the duration of our in vivo experiments (45 days) were much shorter than in the study by Goetz et al., (2011) (75 days). This makes it difficult to interpret the difference between the studies as it is possible that the cells required more time to manifest the difference between treatments observed by Goetz et al. We also found a statistically significant negative correlation of intratumoral remodeling with metastatic burden, while the original study found a statistically significant positive correlation (Figure 7Cd; Goetz et al., 2011), but again there were differences between the studies in terms of the duration of the metastasis studies and the imaging approaches that could have impacted the outcomes. Finally, we report meta-analyses for each result.

Beyond this change to the Abstract, the only other changes that are required are minor changes to the text of the manuscript to address the specific points raised by reviewer #5 below. The comments from reviewer #1 are included below to provide context but would be entirely addressed by the change in the Abstract noted above.

*Reviewer #1:*

I regret to note that the authors have missed a key point that required their attention, even though it was explicitly stated in my previous comments and my guiding suggestions on the lines of change I and other reviewers considered appropriate.

I quote here again the point I aimed at getting across: […] We cannot help noticing the authors keep explicitly stating and ´presenting´ certain divergent observations as results worth being considered […]

This clearly referred to two sentences in the Abstract, which were still included in that previous revision and in the latest version of the manuscript:

"[…] We found metastatic burden was similar between Cav1WT and Cav1KO pMEFs, while the original study found increased metastases with Cav1WT (Figure 7C; Goetz et al., 2011). We also found a statistically significant negative correlation of intratumoral remodeling with metastatic burden, while the original study found a statistically significant positive correlation (Figure 7Cd; Goetz et al., 2011) […]."

I contend again that these parts of their replication study cannot be subject to fair comparison, and should therefore not be presented upfront in this way, equaling them to the rest of the observations, as if they were to be considered as data obtained thoursough appropriate experimental replication. The way the Abstract keeps being presented is misleading (perhaps unintentionally), and leaves room for the wrong interpretation that those in vivo experiments deserve being considered as valid experiments that 'may have yielded a different outcome' for this reason or another. This is not useful for the readership of *eLife* and is damaging to the general aim of this reproducibility initiative, and I respectfully suggest again modifying this text following this example:

"[…] Two key experimental parameters (timing of in vivo metastasis experiments and image-based analysis of tumor ECM architecture) did not match those used in the original study, rendering some observations, such as metastatic burden or its correlation with intratumoral ECM remodeling, not suitable for comparison […]".

I did note and acknowledge that certain changes had partly been made in a good direction, such as the sentence at the end of the Abstract the authors bring up in their response, where they admit that key experimental aspects had not been reproduced and "could have impacted the outcomes [sic]". However, and notwithstanding the previous key point, this statement should at the very least be written before listing those experiments they performed under different conditions.

I am compelled to state again that these details are essential to preserve the original aim of the reproducibility initiative and provide a fair and non-misleading message to its readership. It must be noted this was a common claim of reviewers 2, 4 and us, and should therefore be fully complied with before publication.

*Reviewer #5:*

The authors have now included a specific justification for the humane endpoint, and it gets clear that the procedure was adequate.

The discussion on the implications of shorting the observation period and incomplete reporting of primary tumor burden at the endpoint in the original study could still be discussed with more care.

Specific points:

1) The authors state in subsection “Subcutaneous tumorigenicity assay of tumor cells co-injected with Cav1WT or Cav1KO primary MEFs”: "Although experimental timing is important, maintaining it might not be sufficient to observe the same malignant progression between studies." – it is not clear what is meant with "maintain timing" – the same follow-up period until the metastatic endpoint? The authors should acknowledge that reproducing the timelines of primary tumor growth and spontaneous metastasis as exactly as possible is critical for minimizing confounding parameters, because the multi-step cascade to metastasis including invasion, organ colonization and outgrowth are strongly time-dependent processes. In addition, growth of metastases is nonlinear, i.e. a few days towards later time points might impose major differences in aggregate tumor burden. This should be discussed in more detail.

2) In addition, it would be important to provide a recommendation how such inconsistencies can be mitigated in future work, for example by (i) reporting the tumor volumes for each animal at the endpoint, (ii) resecting the primary tumor by a standardized measure (e.g., size or time), to allow for a follow-up of metastasis development. This would allow to monitor metastasis independent of primary tumor load and premature humane endpoint because of variations in growth of the primary.

3) The readability of labels in Figures 2A, C and Figure 3C remains unacceptable and should be improved.

---

## [Author Response]

Summary:This Replication Study has reproduced some parts of the original paper, but for other parts no statements about reproducibility can be drawn, since the experimental setup was different from the original study.Essential revisions:

*1) The original authors carried out* in vivo *experiments for 70 days, but this replication study terminated them at 45 days. While the rationale for shortening experiments by 25 days (a net reduction of 36% from original conditions!) is respectable, this makes it impossible to draw any firm conclusion with respect to the reproducibility of the original claims related to metastasis, since metastasis is a very time-dependent process. You need to more clearly acknowledge that you cut the metastasis assay short relative to the original study and therefore cannot draw a conclusion concerning the reproducibility of the metastasis results. This should be done in the Abstract.*

We have revised the Abstract to state this key difference.

2) You DID find similar SMA expression levels across genotypes when cultured in 2D plastic. The description of these experiments (subsection “Isolation and characterization of Cav1 wild-type and Cav1 knockout primary MEFs”) is misleading. The observations were not unexpected: the original study had already compared SMA expression between WT and Cav1KO MEFs in 2D and 3D conditions (Figure S2A, original study).

We have revised this description to reflect this comment and underscore the ‘subtle differences’ between WT and KO pMEFs that were communicated to us by the original authors when we sought feedback about this result. Importantly, this experiment was suggested during peer review of the Registered Report as a method to examine ECM remodeling capabilities in vitro, before using the pMEFs for the in vivo experiment. Thus, we were not expecting the result we observed. However, it was communicated to us that a number of factors can influence whether a difference will be observable (e.g. plastic, time after isolation, 2D vs 3D, etc), which we included in this manuscript. And as described in subsection “Deviations from Registered Report”, a recommendation to conduct a collagen contraction assay was agreed with the original authors and pre-registered before experimental work began.

3) The data in Figure 3 exhibit poor staining quality and an overcompensation in digital levels. Hence, the quantitation is not comparable to that of the original study, and this should clearly be noted in the manuscript.

We are unsure what evidence was used by the reviewer to make the statement that the staining was of poor quality. The staining specificity, using two negative controls (no primary antibody or isotype control antibody) was reported in this replication attempt (Figure 3—figure supplement 1), although these staining controls do not appear to have been reported in the original study. This comment may be referring to image noise, the random fluctuation of light intensity contained in images, which is a common issue that affects the quality of fluorescence-based images because of the low-light nature of the signal. In microscopy, noise is produced by the sensor and circuitry of the detectors used (e.g. camera and PMTs) and the unavoidable shot noise which relates to the amount of photons reaching the detector. Since fluorescence imaging demands a sufficient signal-to-noise ratio (SNR) to detect the features of interest, noise can have a negative impact on the subsequently process and analyze steps. Therefore, it is best to adopt strategies to minimize noise, with management steps usually in the image process and analysis protocol. As such, there are several methods to measure noise from either CCD cameras or confocal PMT detectors (Heintzmann et al., 2018,, van Vliet, Sudar and Young, 1998, Murray, 2007). Noise can be measured for particular instruments or from images themselves. Unfortunately, the original study, like many scientific reports publishing fluorescence microscopy images, did not mention any measurements of noise or SNR. Additionally, the original study did not report a strategy used to minimize noise and the original microscopy raw images were not available, therefore we did not have the possibility to compare the amount of noise of the original study verse this replication attempt.

We also disagree that overcompensation in digital levels occurred since there was no overcompensation done as stated in the figure legends (the images for each staining are displayed in the same range of grey levels).

4) Clearly discuss the problem with the primary fibroblasts used in this study, which appear to be less contractile (see Figure 1C-E) than in the initial study (see Figure 1F). The issue is that these fibroblasts could not be further characterized because of antibody cross reactions during SMA staining (Figure 3). In the absence of further characterization of the fibroblasts used in this study, the authors conclusions are undermined.

We disagree that the primary fibroblasts used in this study were any less contractile than the original study. First, the original study figure (Figure 1F) that is referenced did not use pMEFs, but immortalized MEFs. The assay that tested pMEFs in the original study was not reported. Second, at the two time points we analyzed (24 hours and 48 hours after plating), the contraction index reported in Figure 1F of the original study for MEFs (WT MEFs: 24 hours = ~70% and 48 hours = ~82%; KO MEFs: 24 hours = ~55% and 48 hours = ~63%) are quite similar to what we reported in Figure 1D for pMEFs (WT MEFs: 24 hours = 75% and 48 hours = 84%; KO MEFs: 24 hours = 56% and 48 hours = 63%).

We have revised the manuscript to further discuss the inability to further characterize the fibroblasts used in this study due to antibody cross reactions during SMA staining. It was unclear whether this was encountered in the original study and if so, what was performed to deal with the issue. As such, we emailed the original authors when we observed the cross reactions with SMA staining and confirmed the protocol we followed, that was described in the Registered Report, was the same that was performed in the original study. We also highlight the impact this unexpected complication has on interpretation of the results considering the absence of further characterization of the pMEFs.

5) Acknowledge and discuss the fact that not using the specific steps of MetaMorph analysis to score fibronectin fiber orientation could have altered the results.

We agree this could have altered the results obtained and in the previous version of this manuscript included this point, among others, in the final paragraph of the Discussion section. To further highlight this, we have included this in the Abstract and the relevant Results section.

Reviewer #1:The goal of this Reproducibility Project is undermined because differential results are derived from experiments failing to reproduce key aspects of the original study, thus unsuitable for fair comparison. Furthermore, a bias is put into highlighting such 'differences', belittling observations faithfully replicating conditions and yielding results similar to those of the original study (SMA expression levels; collagen contraction assays; even trends for lower ECM fiber order in tumors bearing Cav1KO fibroblasts despite substantial shortcomings).*Goetz et al., carried out* in vivo *experiments for 70 days, but this replication study terminated them at 45 days. While the rationale for shortening experiments by 25 days (a net reduction of 36% from original conditions!) is respectable, this fully invalidates this section to provide a fair assessment of reproducibility of the original paper. Parameters hinting at how limited comparison is include different organ distribution of metastasis, and the failure to detect their luminescence* in vivo. Such a key difference as experiment duration overrules other argued potential sources of variation (see subsection “Subcutaneous tumorigenicity assay of tumor cells co-injected with Cav1WT or Cav1KO primary MEFs"). Metastasis is a non-linear event, and conclusions cannot be drawn if timespan allowed for growth and progression is drastically different.

*Therefore, the authors should acknowledge that* in vivo *experiments simply cannot be compared. The authors briefly mention these pivotal pitfalls somewhere in the text themselves (see for example subsection “Meta-analyses of original and replication effects” and subsection “Subcutaneous tumorigenicity assay of tumor cells co-injected with Cav1WT or Cav1KO primary MEFs"), but nonetheless highlight such differential observations (even in the Abstract!) as relevant, and compare their potential interpretation. It should be explicitly stated in the Abstract, and clarified in the main text, that these sections are not valid attempts to reproduce the original report. I would even suggest moving upfront passages such as those listed above (subsection “Meta-analyses of original and replication effects” and subsection “Subcutaneous tumorigenicity assay of tumor cells co-injected with Cav1WT or Cav1KO primary MEFs"), as they address the core aim of the replication study: to faithfully reproduce original experimental conditions.*

We revised the manuscript to address this comment as addressed above.

Other sections of the reproducibility study that are misleading, or for which clear limitations should be explicitly acknowledged, are the following:First: the authors DID recapitulate similar SMA expression levels between genotypes when cultured in 2D plastic. The description of these incomplete experiments (subsection “Isolation and characterization of Cav1 wild-type and Cav1 knockout primary MEFs”) is however inaccurate and misleading. Their observations were NOT unexpected: the original study had already compared SMA expression between WT and Cav1KO MEFs in 2D and 3D conditions (Figure S2A, original study). The replication study would have benefited from performing CDM-based experiments-which is how Goetz et al., uncovered the reported changes-in addition to collagen contraction assays.

We revised the manuscript to address this comment as addressed above. Regarding the last statement, we appreciate this point of view and also appreciate the original authors informing us about the collagen contraction assays in light of the data we observed during the course of the replication experimentation. However, if performing CDM-based experiments, or collagen contraction assays, to identify pMEFS that obtained a specific outcome in vitro, before using the pMEFs for the in vivo experiment, this replication attempt would have benefited from having this shared during preparation and/or review of the Registered Report. This type of feedback before conducting experiments helps minimize confirmation bias and to maximize the quality of the methodology in an attempt to have no reason to expect *a priori* a different result than the original study.

Second, experiments in Figure 3 exhibit poor staining quality and an over compensation in digital levels. Hence, their quantitation is simply not comparable to that of the original study (where additional complementary techniques were used, such as SHG microscopy). Setbacks when trying to use the original image analysis tools (based on MetaMorph software) justify their using a completely different image setting and analysis approach. While similar trends to the observations from Goetz et al. were recorded, these facts (which add up to the dismal difference in the duration of the experiments) again preclude a fair comparison between both studies.

We responded to this comment above and do not understand what evidence was used by the reviewer to make the statement that the staining was of poor quality. We also acknowledge that the original study used additional complementary techniques, such as SHG microscopy; however, importantly, these were not utilized for the experiment that was replicated and thus are not directly comparable.

In summary, this report unintentionally distorts the purpose of the Reproducibility Project itself, because its experimental execution, interpretation and writing-up are biased, emphasizing differential results despite being irrelevant and not comparable. A major revision of the text should therefore be carried out at the least.Reviewer #2:In this replication study, the authors have attempted to validate some of the original findings published in 2011 by the Del Pozo laboratory (Goetz et al., 2011). Based on previous exchange with the reviewers on one hand, and the authors of the original study on the other hand, it was agreed that two experiments of the original study (mainly Figure 7) were worth replicating: it involved mainly a subcutaneous xenograft of breast cancer derived lung metastatic cells in mice where tumor growth and metastasis would be analyzed.In present study, some of the findings that were originally published by Goetz et al. could be reproduced. First, they could confirm that primary mouse embryonic fibroblasts (pMEF) that express Cav1 have increased extracellular matrix remodeling capability both in vivo and in vitro. They also found in agreement with the initial study that primary growth of the tumor was not affected by the presence of either Cav1 WT or Cav1 KO pMEFs. Unfortunately, they could not reproduce a key result of the original study that is that metastatic dissemination was increased by fibroblasts expressing Cav1 WT. Indeed, no difference could be found here between Cav1 WT and CAV1 KO pMEFs.This is the most annoying result of this replication study as the controversy that exists about Cav1 and fibroblasts deals mainly with the role of Cav1 in metastasis. From the present study, it appears however that the exact experimental conditions of replication could not be respected, and that is certainly the main explanation for this discrepancy. I see two main problems with the replicated experiments. First, the primary fibroblasts used in this study appear to be less contractile (see Figure 1C-E) than in the initial study (see Figure 1F). This is certainly an important issue as the study examines the effect of these fibroblasts on tumor growth and metastasis. I am not convinced that the difference that is shown on Figure 1 C-E is significant. Another issue is that these fibroblasts could not be further characterized because of antibody cross reactions on SMA staining (Figure 3). I was anticipating these problems and this is why I had recommended in my initial review to also examine the involvement of p190RhoGAP in this process. For me in the absence of further characterization of the fibroblasts used in this study, the authors cannot conclude.

As stated above, we disagree that the primary fibroblasts used in this study were any less contractile than the original study. Nonetheless we agree that there might be some ‘minimal’ effect size of in vitro contractility that might be needed to observe an in vivo effect. Importantly, observing different outcomes are informative to establish a range of conditions under which a given effect can be observed, a point we’ve included in the revised manuscript. Additionally, we appreciate the view that additional information (i.e. examining p190RhoGAP) would be beneficial, and have included this in the revised manuscript as well.

The most important issue with this study is the considerable difference in the time of observation of metastatic foci in various organs. While the original study observed metastasis after 70 days, here the observation is done at a considerably earlier time i.e. 45 days that is almost half the time. Considering that metastasis is a secondary event that takes time to occur, it is likely that the observation has been done too prematurely.

We have revised the Abstract to state this key difference

In conclusion, the replication study shows several similar aspects with the original study by Goetz et al., regarding the role of Cav1 WT expressing cancer associated fibroblasts in remodeling the extracellular matrix although it could not be statistically validated. The lack of reproducibility for the metastasis process is most likely due to the impossibility to respect the same experimental protocol than the one in Goetz et al. Combined with the fact that the fibroblasts injected here present a weaker contractile activity, and thus may also explain the absence of effect in metastasis. This is clearly an experimental weakness of this study.In the Discussion section, the authors should discuss more these two important discrepancies rather than involving a list of experimental features (subsection “Meta-analyses of original and replication effects”), which if it is true that they may be involved would then be always present in all studies and as a matter of fact would prevent from reproducing any experiment. Also, the possibility that human cancer cells (the ones injected here) may behave differently in a mouse model should be discussed as it could explain the discrepancies observed with other studies where mouse cancer cells were injected in mice (see Capozza reference cited in the original study and not cited here).

We have expanded the discussion on these two differences as well as the additional factor raised. Regarding the point of differences and their impact on replicating experiments, yes, it is possible any of these (among many others that can be hypothesized) could influence the outcome, which is true for any experiment. But whether they actually are is the open question. Replication plays a role here to test what is thought to matter, that is, this replication attempted to reproduce a previous finding with no *a priori* reason to expect a different outcome. When a different result is obtained, there are many possible factors that could explain the differences. Conversely, because it’s not possible to do the exact same experiment again, since there will always be a difference between the original and replication studies, when a similar result is observed in a replication, despite the known and unknown differences between studies, it increases confidence in the original finding as well as the generalizability of the result. This paper offers further insight on this topic: Errington and Nosek, 2017.

Reviewer #4:

The goal of this manuscript to replicate a cancer biology study published in Cell appears well-intentioned and potentially valuable. Unfortunately, the authors make two major changes to the original study protocol that have unknown and potentially substantial consequences, making it impossible in my opinion to provide a valid replication test of the original study – even though multiple other aspects of the approach such as pre-registration and the careful attempts to identify specific reagents and materials were commendable.1) The first major change, even though seemingly within the parameters of the loosely written original plan that allowed for adjusting study length, has potential major problems. In any spontaneous metastasis assay, extending the time for analysis by 55% (25 days beyond 45 days to reach the 70 days in the original paper) could have major effects on both the extent and patterns of metastases. The current authors used a markedly truncated experimental endpoint for all animals even if many may well have not been in distress – the result is that comparisons of results between the two studies performed at 45 days versus 70 days seems impossible. Effects seen in the original study may have been missed or may not, which is impossible to determine. In fact, if the original Registered Report had explicitly proposed substituting an experimental endpoint for spontaneous metastasis of 45 days in place of the original 70 days, it seems unlikely that it would have been approved.

We appreciate this perspective and have revised the manuscript to address this comment as discussed above.

2) This major change in experimental endpoint timing also likely compromised the comparisons of fibronectin fiber patterns. Although it is impossible to know, it is plausible that extending the time of tumor interactions with the local microenvironment by 55% might affect the fibronectin pattern associated with intratumoral remodeling. Unless the original conditions could be met at least approximately, it seems like comparing apples and oranges (or green vs. over-ripe apples).

We agree and have revised the manuscript to reflect this.

3) A second potentially important change was complete substitution of alternative image analysis methods concerning fibronectin patterns compared to the Registered Report describing the use of MetaMorph and scoring SMA-positive cells. It is simply not clear whether scoring all cells rather than SMA-positive cells could have affected the results. More importantly, not using the specific steps of MetaMorph analysis to score fibronectin fiber orientation could have altered the results. The authors provide somewhat plausible reasons for not scoring SMA-positive cells (the staining did not work in their hands) and avoiding the use of MetaMorph (they found "too many objects to process") but changing the workflow completely appears to invalidate the attempted replication. Unless the current authors can demonstrate that their altered methods provide the same outputs on the original data as the original methods [e.g., by obtaining primary data from the original authors], these substitutions involving altering the cell population analyzed (all cells vs. SMA+) and image analysis (KNIME analytics vs. MetaMorph) fail to provide direct experimental replication. In fact, a major specific defect of the new analysis is apparent from Figure 3—figure supplement 3 in which the software fails to capture the fibrillar nature of fibronectin staining in 4 out of 9 samples in which average brightness is merely lower: the authors show dots and short lines rather than the interconnected fibronectin fibrillar staining patterns.

We agree that this was a challenging aspect of this replication attempt. Regarding the first point about SMA staining and the comparison of SMA-positive cells. The analysis conducted in the original study (in Figure S7Cc) and outlined in the Registered Report (Protocol 4) described two analyses on fibronectin orientation. One using all cells, which we report here, and one on just SMA-positive cells, which we were unable to conduct due to challenges we could not overcome with the SMA staining procedure used. Thus, we did not change the population of cells being investigated, but instead could not conduct a sub-analysis on just the SMA-positive cells as planned.

We agree that a change in workflow can have unexpected and not easily comparable aspects to it for assessing reproducibility. The suggestion to compare the workflow we generated on original primary data was something we explored, but we were unable to obtain original primary data or macros to do this (as a note, we asked this during the Registered Report because even details of the MetaMorph protocol were not explicit in all aspects). While we can not provide a benchmark for how the workflow we had to implement compares to the original study protocol, we were encouraged that the overall median values of percent of fibronectin fibers oriented with 20% we observed [*Mdn*=42.1%, IQR=39.1-44.7%, n=20] were generally similar to the original values [*Mdn*=42.5%, IQR=37.9-45.1%, n=23]. And yes, we agree there is the possibility of additional optimization of the analysis we present, especially as the reviewer points out variations in signal intensity between images. Importantly, though it is unclear how variable the original images were or how the original study handled these variations in their analysis pipeline as described. Importantly, the approach should be done blinded to the conditions and batch processed to mitigate any potential for bias. We also conducted additional complementary analysis procedures using FibrilTool (as recommended by the original authors during preparation of the Registered Report as an alternative approach to perform the analysis), and OrientationJ. Although the full range of possible methods were not explored, we found the results of all these methods, plus a blinded manual scoring, were reasonably well correlated indicating the robustness of the findings presented. These points are further discussed in the revised manuscript.

4) Although not the approach taking by these well-intentioned replicators, this reviewer believes that more direct exchanges of expertise and experience between the original research group and the replication group could have avoided the serious problems listed above that, at present, unfortunately invalidate this well-intentioned replication attempt.

We appreciate this perspective from the reviewer and do not disagree with it in principle. To this point, we made much effort to do this during the course of this study and appreciated the immense effort of the original authors to help us both in preparation of the Registered Report, but also during the experimentation. However, it is also valuable to know under what circumstances more direct exchanges are needed and how those exchanges take place. We argue that increased transparency of process *and* outcome is important to mitigate these concerns, hence the approach we took with this project (see: Errington et al., 2014). We also recognize that in some circumstances further exchanges might be beneficial. However, this is not feasible or practical to do this for all replication attempts.

[Editors' note: further revisions were requested prior to acceptance, as described below.]

The manuscript has been improved but there are some remaining issues that need to be addressed before acceptance. The strong consensus among the referees and reviewing editor is that the shortening of the metastasis period makes it difficult to compare this study with the original one, and the manner in which this point is presented in the revised manuscript is not satisfactory. In particular, you must include additional discussion on your understanding of the relevance of a shorter or longer followup period for metastasis outcome, and the manner in which variations in this important parameter may influence the outcome of the experiments and, with that, the robustness of the conclusions that can be drawn in your study about the reproducibility of the original results.

We have included further discussion on the implications of a changed time course in metastatic studies in the revised manuscript and the impact this could have when comparing the two studies.

Reviewer #1:Fiering et al., complied in this revised version with several previous recommendations concerning the summary text, as well as changes to the main text body of the manuscript. However, the rewritting of the Abstract, while in the requested direction, should still be improved. We cannot help to notice the authors keep explicitly stating and ´presenting´ certain divergent observations as results worth being considered:"[…] We found metastatic burden was similar between Cav1WT and Cav1KO pMEFs, while the original study found increased metastases with Cav1WT (Figure 7C; Goetz et al., 2011). We also found a statistically significant negative correlation of intratumoral remodeling with metastatic burden, while the original study found a statistically significant positive correlation (Figure 7CD; Goetz et al., 2011). Finally, we report meta-analyses for each result."*As they subsequently state, these differences can be explained alone by the fact that key experimental details were not reproduced (i.e. the replication study shortened the duration of* in vivo experiments by more than 35% as compared to the original study, and they used a different image approach when assessing ECM architecture from microscopy images (apart from additional technical issues, already mentioned in the previous round of review)). Thus, we claim again that these parts of their replication study cannot be subject to fair comparison as such and should not be presented upfront with the same value as the rest of the observations. Moreover, the fact that key experimental aspects were not fully reproduced should be clearly stated before those descriptions. To provide an example of what, in our opinion, should be stated in the Abstract regarding these experiments:
*"[…] Two key experimental parameters (experimental timing of* in vivo metastasis experiments and image-based analysis of tumoral ECM architecture) did not match with those used in the original study, rendering some observations, such as metastatic burden or its correlation with intratumoral ECM remodeling, not suitable for comparison. Finally, we report meta-analyses for each result."While we appreciate that an effort has been made trying to correct these issues according to the previous revision round, we think these details are paramount to avoid any ambiguous or misleading messages in the Abstract to potential readers. Besides, this was a common claim of reviewers 2, 4 and us (while the only other referee focuses on the statistical analysis and presentation of the data).

The quoted text of our manuscript Abstract above is from the first version of the manuscript and does not include the revisions that were made. Following the last round of peer review, we included an additional sentence that specific factors (experimental timing and image analysis approach) could have impacted the outcomes.

Reviewer #5:

*This replication study addressed experiments performed by Goetz et al., (2011) which indicated a critical role of caveolin-1 expressed by myofibroblasts of the tumor stroma in ECM remodeling and its role in inducing distant metastasis formation. The results confirm the originally shown role of caveolin-1 in collagen contraction and, to some extent, in fibronectin matrix alignment* in vivo*, as well as a lack of involvement in tumor growth at the implantation site. Conversely, this replication study could replicate the effect of caveolin-1 positive fibroblasts on metastatic evasion, and even indicated an inverse effect between metastasis outcome and fibronectin alignment in the primary tumor.*

With the notable exception of a reduced followup period of metastasis after tumor implantation. The experiments were designed and performed with high fidelity and the quality of documentation is excellent. The meta-analysis further explores similarities and differences with competence and is delineated in a comprehensive manner. As major shortcoming, the explanation justifying the shortened observation period for analyzing metastasis outcome and the discussion on the implications for similar replication work in general are not yet satisfactory.

We have included additional information in the body of the text for the shortened observation period in the revised manuscript, which is addressed in the specific point about subsection “Subcutaneous tumorigenicity assay of tumor cells co-injected with Cav1WT or Cav1KO primary MEFs” below. Additionally, we have included further discussion on the implications of a changed time course in metastatic studies.

Specific points:Subsection “Subcutaneous tumorigenicity assay of tumor cells co-injected with Cav1WT or Cav1KO primary MEFs": "45 days after cell injection to maximize the length of time for tumor growth while minimizing animal suffering" – it should be stated whether in this replication study, the tumors grew more rapidly, compared to Goetz et al., and whether a different tumor size until human endpoint mandated this deviation compared to the original study.

It was not reported in the original paper, or shared with us, what the tumor growth rates were for the original study. However, the shortened time course followed in this replication, to minimize animal suffering, suggests the possibility of a faster tumor growth rate observed in this replication than the original study. This section of the manuscript has been revised to reflect this.

For a general audience, the authors should include additional discussion on their understanding of the relevance of a shorter or longer followup period for metastasis outcome. Particularly, they should discuss how robust results can be considered and how similar replication work in independent labs should be performed. Is it legitimate to adjust followup periods? Which rules should apply to a robust biological outcome? How can the inverse correlation of metastasis and ECM remodeling in the primary tumor site be explained when compared to the primary work? Which growth monitoring criteria should be applied to achieve high fidelity replication? How should researchers deal with differences in human endpoint criteria when constrained by a given legal framework? How would the authors perform this part of the work in retrospect, to achieve higher concordance with the original study?

We have included additional discussion on the impact of experimental time for metastasis outcome. While we addressed some of the specific questions raised, not all were addressed as we think they would be better addressed in an insight to this paper or in a global assessment of all replications that were conducted as part of the Reproducibility Project: Cancer Biology, not just this single replication.

The discussion on intermittent parameters potentially affecting the bioluminescence imaging is of general interest. It is not clear, however, whether these points are relevant here, assuming that the same imaging approach was used as in Goetz et al. (whole-body bioluminescence). However, Goetz et al., additionally used bioluminescence analysis of excised organs. Can the authors clarify why this more sensitive approach was apparently not used here?

The same approach as Goetz et al., was used in this replication. As described in subsection “Subcutaneous tumorigenicity assay of tumor cells co-injected with Cav1WT or Cav1KO primary MEFs” and the methods section ‘subcutaneous tumorigenicity assay’, tumors were imaged by whole-body bioluminescence (measurements reported in Figure 2A with representative images in lower panel of Figure 2B) and then organs and tumors were excised and imaged ex vivo (measurements reported in Figure 2C with representative images in upper panel of Figure 2B).

[Editors' note: further revisions were requested prior to acceptance, as described below.]

The manuscript has been improved but there are a few remaining issues that need to be addressed before acceptance, as outlined below:1) The most critical issue is that the editors and reviewers were not satisfied with the clarity of the Abstract in terms of the way it acknowledges the potential impact of the shorter metastasis assay on the interpretability of the results. Based on extensive consultation among the editors and reviewers, their concerns would be resolved if you would be willing to replace the last four sentences in the Abstract with the following text:*We found metastatic burden was similar between Cav1WT and Cav1KO pMEFs, while the original study found increased metastases with Cav1WT (Figure 7C; Goetz* et al.*, 2011); however, the duration of our* in vivo experiments (45 days) were much shorter than in the study by Goetz et al., (2011) (75 days). This makes it difficult to interpret the difference between the studies as it is possible that the cells required more time to manifest the difference between treatments observed by Goetz et al. We also found a statistically significant negative correlation of intratumoral remodeling with metastatic burden, while the original study found a statistically significant positive correlation (Figure 7Cd; Goetz et al., 2011), but again there were differences between the studies in terms of the duration of the metastasis studies and the imaging approaches that could have impacted the outcomes. Finally, we report meta-analyses for each result.Beyond this change to the Abstract, the only other changes that are required are minor changes to the text of the manuscript to address the specific points raised by reviewer #5 below. The comments from reviewer #1 are included below to provide context but would be entirely addressed by the change in the Abstract noted above.

We have revised the Abstract as suggested and addressed the specific points raised below by reviewer #5.

Reviewer #1:I regret to note that the authors have missed a key point that required their attention, even though it was explicitly stated in my previous comments and my guiding suggestions on the lines of change I and other reviewers considered appropriate.I quote here again the point I aimed at getting across: […] We cannot help noticing the authors keep explicitly stating and ´presenting´ certain divergent observations as results worth being considered […]This clearly referred to two sentences in the Abstract, which were still included in that previous revision and in the latest version of the manuscript:"[…] We found metastatic burden was similar between Cav1WT and Cav1KO pMEFs, while the original study found increased metastases with Cav1WT (Figure 7C; Goetz et al., 2011). We also found a statistically significant negative correlation of intratumoral remodeling with metastatic burden, while the original study found a statistically significant positive correlation (Figure 7Cd; Goetz et al., 2011) […]."*I contend again that these parts of their replication study cannot be subject to fair comparison, and should therefore not be presented upfront in this way, equaling them to the rest of the observations, as if they were to be considered as data obtained thoursough appropriate experimental replication. The way the Abstract keeps being presented is misleading (perhaps unintentionally), and leaves room for the wrong interpretation that those* in vivo experiments deserve being considered as valid experiments that 'may have yielded a different outcome' for this reason or another. This is not useful for the readership of eLife and is damaging to the general aim of this reproducibility initiative, and I respectfully suggest again modifying this text following this example:
*"[…] Two key experimental parameters (timing of* in vivo metastasis experiments and image-based analysis of tumor ECM architecture) did not match those used in the original study, rendering some observations, such as metastatic burden or its correlation with intratumoral ECM remodeling, not suitable for comparison […]".I did note and acknowledge that certain changes had partly been made in a good direction, such as the sentence at the end of the Abstract the authors bring up in their response, where they admit that key experimental aspects had not been reproduced and "could have impacted the outcomes [sic]". However, and notwithstanding the previous key point, this statement should at the very least be written before listing those experiments they performed under different conditions.I am compelled to state again that these details are essential to preserve the original aim of the reproducibility initiative and provide a fair and non-misleading message to its readership. It must be noted this was a common claim of reviewers 2, 4 and us, and should therefore be fully complied with before publication.Reviewer #5:The authors have now included a specific justification for the humane endpoint, and it gets clear that the procedure was adequate.The discussion on the implications of shorting the observation period and incomplete reporting of primary tumor burden at the endpoint in the original study could still be discussed with more care.Specific points:1) The authors state in subsection “Subcutaneous tumorigenicity assay of tumor cells co-injected with Cav1WT or Cav1KO primary MEFs”: "Although experimental timing is important, maintaining it might not be sufficient to observe the same malignant progression between studies." – it is not clear what is meant with "maintain timing" – the same follow-up period until the metastatic endpoint? The authors should acknowledge that reproducing the timelines of primary tumor growth and spontaneous metastasis as exactly as possible is critical for minimizing confounding parameters, because the multi-step cascade to metastasis including invasion, organ colonization and outgrowth are strongly time-dependent processes. In addition, growth of metastases is nonlinear, i.e. a few days towards later time points might impose major differences in aggregate tumor burden. This should be discussed in more detail.

We have revised the text to remove this sentence and to expand the importance of time and non-linear growth.

2) In addition, it would be important to provide a recommendation how such inconsistencies can be mitigated in future work, for example by (i) reporting the tumor volumes for each animal at the endpoint, (ii) resecting the primary tumor by a standardized measure (e.g., size or time), to allow for a follow-up of metastasis development. This would allow to monitor metastasis independent of primary tumor load and premature humane endpoint because of variations in growth of the primary.

We have included additional text to discuss these recommendations in the revised manuscript.

3) The readability of labels in Figure 2A, C and Figure 3C remains unacceptable and should be improved.

We have revised the labs in these figures to increase their readability and to make them consistent with the other figure labels.